# A Slam-dependent hemophore contributes to heme acquisition in the bacterial pathogen *Acinetobacter baumannii*

Thomas J. Bateman[1], Megha Shah[1], Timothy Pham Ho[1], Hyejin Esther Shin[1], Chuxi Pan [1], Greg Harris[2], Jamie E. Fegan[3], Epshita A. Islam[1], Sang Kyun Ahn[3], Yogesh Hooda[1], Scott D. Gray-Owen [3], Wangxue Chen[2] & Trevor F. Moraes [1✉]

Nutrient acquisition systems are often crucial for pathogen growth and survival during infection, and represent attractive therapeutic targets. Here, we study the protein machinery required for heme uptake in the opportunistic pathogen *Acinetobacter baumannii*. We show that the *hemO* locus, which includes a gene encoding the heme-degrading enzyme, is required for high-affinity heme acquisition from hemoglobin and serum albumin. The *hemO* locus includes a gene coding for a heme scavenger (HphA), which is secreted by a Slam protein. Furthermore, heme uptake is dependent on a TonB-dependent receptor (HphR), which is important for survival and/or dissemination into the vasculature in a mouse model of pulmonary infection. Our results indicate that *A. baumannii* uses a two-component receptor system for the acquisition of heme from host heme reservoirs.

[1] Department of Biochemistry, University of Toronto, Toronto, ON, Canada. [2] National Research Council Canada, Human Health Therapeutics (HHT) Research Center, Ottawa, ON, Canada. [3] Department of Molecular Genetics, University of Toronto, Toronto, ON, Canada. ✉email: trevor.moraes@utoronto.ca

*A* cinetobacter baumannii is a Gram-negative bacterial pathogen that causes both hospital and community-acquired infections worldwide[1]. Clinical manifestations include ventilator-associated pneumonia, sepsis, soft tissue infections, meningitis, and urinary tract infections (UTIs). Multidrug-resistant *A. baumannii* is an urgent public health threat and poses a grave risk to patients due to its ability to persist on desiccated surfaces, and its plastic genome giving it the ability to resist conventional and last lines of antibiotic treatments[2]. *A. baumannii* has recently topped the World Health Organization's list of deadly superbugs that require immediate research into the development of novel therapeutics[3]. Despite sustained efforts to understand the biology of this pathogen, much work remains to be done to uncover its virulence factors and mechanisms of pathogenesis.

*A. baumannii*'s demand for iron, which plays a critical role in life-sustaining processes such as DNA replication and metabolism, is a potential weakness that can be exploited for therapeutic intervention. In the context of humans, heme is the largest reservoir of iron, however, it is largely inaccessible as it is tightly bound to large bulky proteins such as hemoglobin (Hb) and albumin[4]. Gram-negative pathogens have adopted different strategies involving specialized protein machinery for pirating heme from these hemoproteins, and transporting heme across the outer membrane (OM). Heme uptake systems rely on β-barrel proteins called TonB-dependent outer membrane receptors (TBDRs), to steal and transport heme across the OM[5]. Accessory proteins also facilitate heme uptake, and include high-affinity secreted heme-binding proteins termed hemophores and surface anchored lipoproteins (SLPs)[5].

A previous bioinformatics study identified two potential gene clusters involved in heme utilization (Fig. 1a)[6]. The first is present in all *A. baumannii* strains, and encodes a predicted TBDR referred to as HemTR[7]. The second predicted heme uptake cluster, also called *hemO* cluster for encoding the heme-degrading enzyme, heme oxygenase, was found in roughly two-thirds of clinical epidemic isolates[6]. Comparison of the hypervirulent LAC-4 strain possessing both clusters with strains lacking the *hemO* cluster revealed a growth advantage for LAC-4 in minimal media with heme as an iron source, and sera likely attributed to the presence of heme in the form of albumin and hemopexin (Hpx)[8]. Increased sensitivity of LAC-4 to the toxic heme analog, gallium protoporphyrin (GaPPIX), which blocked the spread of LAC-4 from the lungs into the blood in a mouse infection model indirectly demonstrated the importance of the *hemO* cluster to heme uptake and the central role of heme in *A. baumannii* disease progression[8]. The presence of heme cluster 2 in an entire panel of *A. baumannii* multidrug-resistant clinical isolates involved in hospital outbreaks also further highlights the importance of this system in the context of disease[9].

The *hemO* cluster contains a gene (*ABUW_2985*) that encodes a unique TBDR referred to as hemophilin receptor or HphR. We discovered that a nearby gene, *ABUW_2983*, which was previously annotated as encoding a TonB protein, is homologous to a surface lipoprotein assembly modulator or Slam[10]. Our lab discovered that Slam proteins function as conduits for the translocation of SLPs to the cell surface, and that a Slam deletion renders a human restricted pathogen avirulent likely due to the absence of SLPs that normally play key roles as virulence factors facilitating nutrient acquisition, host colonization, and immune evasion[10–12]. Adjacent to *A. baumannii* Slam, which we have named HsmA for hemophilin secretion modulator (HsmA), is a gene (*ABUW_2984*) previously annotated to encode a hypothetical protein that we later discovered contains a canonical lipidation motif or lipobox[13], and may be a Slam-dependent SLP substrate. A recent study demonstrated that a homolog of ABUW_2984

referred here as hemophilin or HphA, is a secreted heme-binding protein produced by *A. baumannii* in iron restrictive conditions[14,15].

Here, we show that *A. baumannii* Slam secretes HphA, which functions as a hemophore that acquires heme from hemoglobin and albumin. HphA along with its cognate receptor, HphR, function together as a two-component receptor required for full virulence and facilitating extrapulmonary dissemination and/or survival in the blood.

## Results

**Heme cluster 2 is important for heme uptake from human hemoproteins.** To examine the contributions of each predicted heme utilization cluster for *A. baumannii* growth, iron starved single-knockout mutants (Fig. 1b–e)[16] were grown in an iron-poor media (Supplementary Fig. 1a) supplemented with human hemoproteins as the sole iron source. Serum human hemoglobin levels are estimated to be 80–800 nM[17]. At the higher end of this spectrum, Fig. 1b shows that knockout of genes in heme clusters 1 and 2 does not impair *A. baumannii* growth. These gene clusters appear redundant as a double knockout mutant exhibited clear growth defects in abundant Hb and not in rich media compared to the single mutants (Fig. 1c; Supplementary Fig. 1e, f). Interestingly, heme cluster 2 outer membrane protein machinery (HsmA, HphA, and HphR) is required for growth in sub-nanomolar and micromolar concentrations of hemoglobin and hemin, respectively, while a HemTR mutant is not impaired in growth (Fig. 1d; Supplementary Fig. 1b, c). HphA, HsmA, and HphR also imparts *A. baumannii* with the ability to utilize heme complexed to human serum albumin (HSA) as a heme source (Fig. 1e; Supplementary Fig. 1d). This supports previous research which found that heme cluster 2 increased *A. baumannii*'s sensitivity to toxic heme analogs in the presence of sera[18].

Furthermore, we have narrowed down the minimal components of heme cluster 2 required for heme transport across the OM to four genes using a reconstituted *E. coli* heme uptake system. *E. coli* has the necessary components to utilize heme as an iron source, but lacks the OM transporters to move heme into the periplasm[19]. Expression of HsmA, HphA, HphR, and TonB related protein (ABUW_2982) enables *E. coli* to use hemoglobin as an iron source without conferring a general growth advantage in iron-rich media (Supplementary Fig. 2a, b). Deletion of HphR eliminates growth specific to heme uptake, growing to empty vector background levels providing further support for heme cluster 2 functioning as a high-affinity heme acquisition system.

**HphA is a secreted hemophore that binds and acquires heme from hemoglobin.** We next focused on examining the mechanistic details of heme uptake, specifically characterizing the Slam-HphA system present in heme cluster 2 (Supplementary Fig. 3). We first studied this system in *E. coli* as we, and others have shown that lipoproteins can reach the outer membrane via the localization of lipoprotein (Lol) export pathway[10,20], but are unable to display lipoproteins due to the absence of a Slam homolog. We used the *E. coli* assay as a platform to test whether HphA is a Slam substrate (Supplementary Fig. 4). Co-expression of HphA with HsmA results in HphA cell surface localization. In this assay, we replaced the endogenous HphA signal peptide with the pelB signal sequence for optimal Sec targeting followed by the predicted HphA lipobox motif, and this likely explains the surface anchoring phenomenon we see in *E. coli*. In *A. baumannii*, HphA is released into the supernatant as first reported by Giardina et al. Knockout of HsmA in *A. baumannii* abrogates HphA secretion, which can be rescued by transformation with a plasmid containing

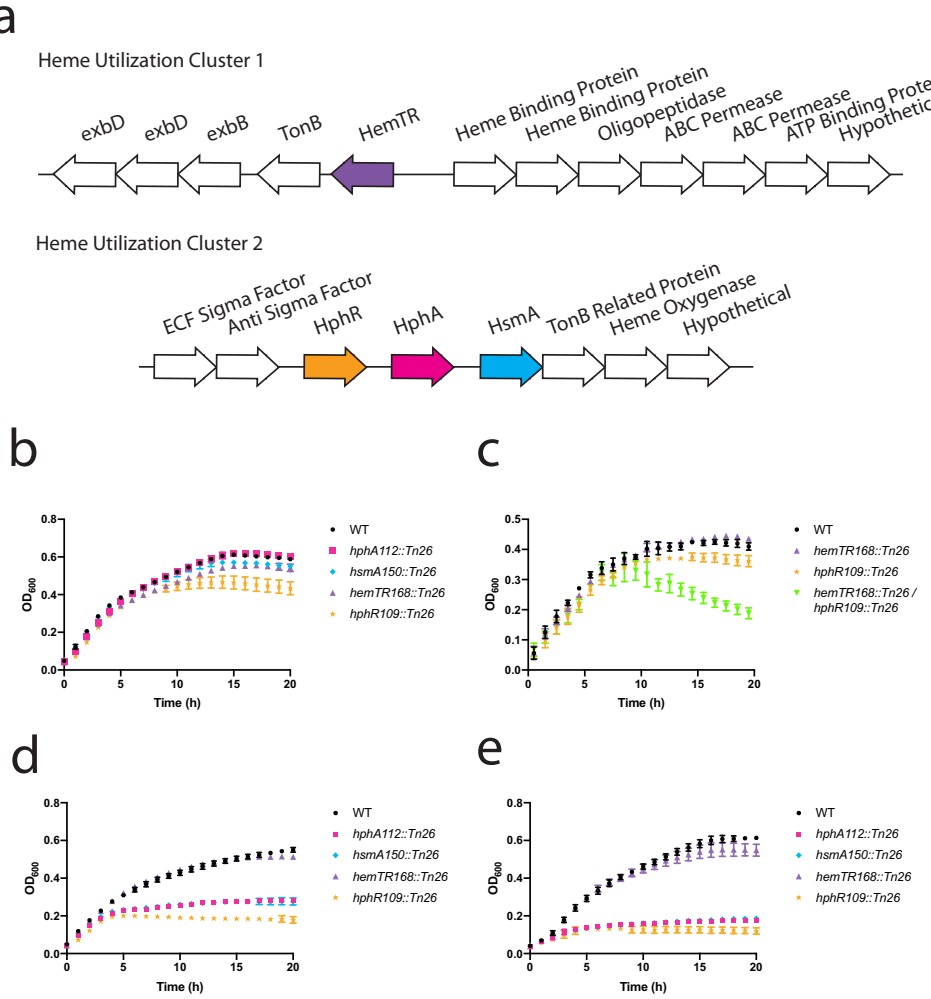

**Fig. 1 Heme cluster 2 enables *Acinetobacter baumannii* to utilize hemoproteins for growth. a** Two putative heme acquisition gene clusters in *A. baumannii*. Annotations of the proteins encoded by these genes are listed above the arrows. Both heme clusters contain a unique TonB-dependent outer membrane receptor (HemTR and HphR). Heme cluster 2 contains a Slam called HsmA adjacent to a predicted substrate of unknown function (HphA). **b–e** To tease apart the roles of the putative clusters in heme-dependent growth, WT, single and double transposon mutant *A. baumannii* (AB5075) strains were iron starved on LB agar supplemented with 200 μM dipyridyl. Cells were inoculated into RPMI supplemented with the following heme sources: 1 μM hemoglobin (**b** and **c**), 75 nM hemoglobin (**d**), and 2 μM hemin with 4 μM human serum albumin (HSA) (**e**). Hemoglobin concentrations expressed in terms of heme equivalents. Growth at 37 °C was monitored with optical density (OD) at 600 nM. Values plotted represent the mean ± standard error of the mean from three independent experiments.

HsmA (Fig. 2a; Supplementary Fig. 12). Together, these results show that HphA secretion is a Slam-dependent process.

We purified and solved the X-ray crystal structure of HphA to gain insight into its role in heme uptake. HphA is composed of a C-terminal 8-stranded β-barrel packed against an N-terminal clamp-like structure through 6 intermediate β-strands (Fig. 2b; Supplementary Table 1). The protein co-purifies with heme which is bound to the N-terminal clamp. The heme-Fe is coordinated by two His residues, H43 and H106, found in a loop and $3_{10}$ helix, respectively (Fig. 2c). The porphyrin ring is oriented in the pocket with both of its propionate groups exposed to solvent, with one forming polar contacts with Y59 and S105. The heme-binding pocket is lined with hydrophobic and aromatic residues contributed by an α-helix and β-strands separating the clamp and barrel. Strikingly, HphA shares the same architecture as the 5 Slam-dependent neisserial surface lipoproteins and a newly discovered secreted hemophore produced by *Haemophilus haemolyticus*[15] (Supplementary Fig. 5). The barrel dominates the similarity between HphA and the neisserial SLPs and hemophore,

while the handle domain of HphA provides the distinct binding machinery for heme sequestration.

To examine the heme-binding potential of HphA, acid-acetone extraction was used to strip the co-purifying heme[21,22]. The resulting apo protein is devoid of heme and refolds to its native conformation as indicated by the absence of a Soret signature in the UV–Vis spectra and X-ray crystal structures (Supplementary Figs. 6a–d, 7; Supplementary Table 1). Comparison of holo and apo structures reveals that the loop containing H43 undergoes a large conformational change upon heme binding and release, while the $3_{10}$ helix containing H106 is only slightly displaced (Supplementary Fig. 7). Apo HphA also crystallized in a $C222_1$ space group (Supplementary Fig. 6c), with three monomers in the asymmetric unit. All apo HphA chains adopt an open conformation and do not show structural heterogeneity in the heme-binding region (Supplementary Fig. 6d). Size exclusion chromatography-multi-angle light scattering (SEC-MALS) confirmed that HphA exists as a monomer in solution, regardless of the presence of heme (Supplementary Fig. 6e).

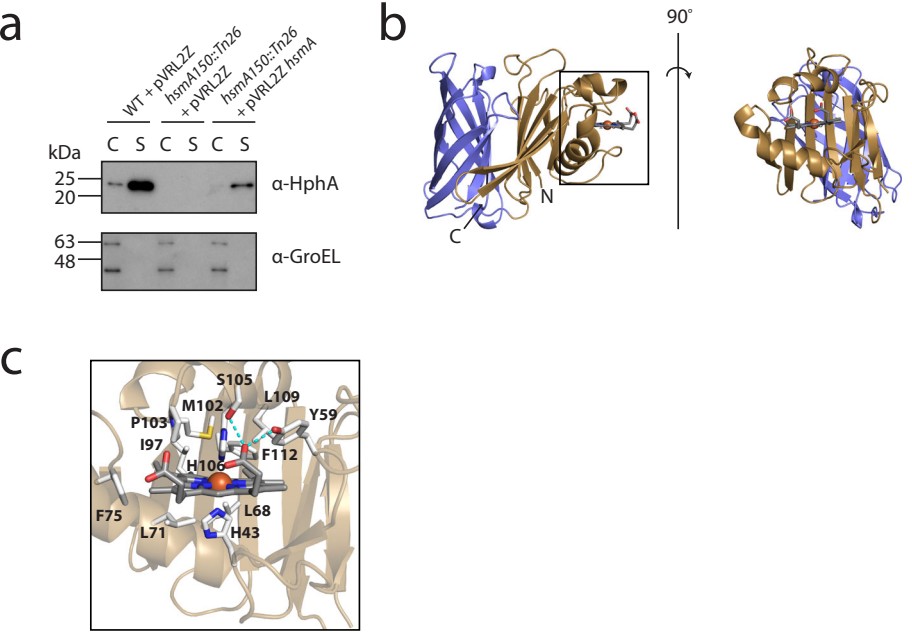

**Fig. 2 HphA is a heme-binding protein dependent on HsmA for secretion. a** Western blots demonstrate that *A. baumannii* secretes HphA into the supernatant, and secretion is HsmA dependent. GroEL is used as a control for cell lysis. Western blots are representative of three independent experiments. C and S refer to cell pellet and supernatant, respectively. **b** Rotated side representations of the structure of holo HphA bound to a molecule of heme. **c** Detailed view of the heme-binding site. Residue side chains in close proximity to heme are depicted as sticks, and polar contacts in cyan.

With over 95% of heme bound to hemoproteins and 67% of heme bound to hemoglobin[17], we sought to test if HphA could bind to and extract heme from hemoglobin. Apo HphA interacts specifically with hemoglobin conjugated to agarose beads as evidenced by pull-down (Fig. 3a, b; Supplementary Fig. 8a). However, heme-loaded WT HphA or the H43A/H106A double mutant that does not co-purify with heme show no or reduced binding to Hb, respectively. To rule out that the interaction is an artefact of using Hb resin, we performed the reciprocal pull-down (Supplementary Fig. 8b). Apo Hph-GST immobilized to glutathione beads binds to Hb and this is not attributed to the GST tag, providing further support that Hph and Hb are interacting partners. The interaction between HphA and hemoglobin is not dependent on the heme content of hemoglobin as evidenced by pull-down, which shows robust binding of apo HphA and residual binding of holo and the heme nonbinding mutant to apo hemoglobin resin (Supplementary Fig. 9). The visible spectrum of unbound apo-HphA after separation from holo hemoglobin beads confirmed the presence of a Soret signature indicative of heme piracy (Fig. 3c). Virtually no Soret or free heme signatures were detected in the double His mutant illustrating its lack of heme-binding activity. Hpx is a known heme scavenger and was also found to acquire heme from Hb resin in our experimental conditions implying that HphA steals heme through a passive process likely by virtue of its high affinity for heme (Fig. 3d). HphA binding to Hb also does not appear to stimulate heme release over time as its scavenging activity is comparable to Hpx (Fig. 3d).

To compare the biological activities of WT and mutant HphA, purified proteins were added to WT and mutant *A. baumannii* growth media. As expected, holo HphA completely rescues growth of the *A. baumannii* HsmA mutant and is unable to support growth of the HphR mutant (Fig. 4a). Partial rescue of the HphA mutant is suggestive of polar effects on promoter-distal genes, which we confirmed using mutants complemented with an *A. baumannii* specific expression vector (Supplementary Fig. 10a–c). Plasmid complementation of the HphA mutant does not rescue growth with Hb, and although not statistically significant, weakly dampens growth in the presence of purified holo HphA. This slight toxicity may be

due to polar effects on downstream HsmA expression causing HphA to accumulate intracellularly and restrict *A. baumannii* access to heme. This explanation is consistent with the appearance of a low molecular weight band, which could be a potential HphA degradation product in the HphA complemented mutant (Supplementary Fig. 10d). We also verified the other transposon mutants and found that plasmid complementation of the HsmA mutant restores growth in the presence of hemoglobin (Supplementary Fig. 10a). Western blot analysis showed reduced HphA expression in the HphR mutant which explains why growth can only be fully restored by providing holo HphA to the complemented mutant (Supplementary Fig. 10b, d). Purified apo HphA does not support growth of HsmA, HphA, or HphR mutants, but rescues growth of the HsmA and HphA mutant in combination with hemoglobin as the heme source (Figs. 4b; 5a–f). Likewise, purified H43A/H106A HphA does not serve as an heme/iron source, and cannot scavenge heme from Hb consistent with its inability to bind heme (Figs. 4c; 5a–f).

**Heme uptake via cluster 2 is important for virulence and systemic spread.** To assess the relevance of heme utilization cluster 2 to bacterial fitness and virulence in vivo, two mice infection models were performed. We grew bacteria in iron restrictive conditions prior to challenge experiments in order to deplete intracellular reserves and force the bacteria to rely on their iron acquisition systems. However, iron starvation did not attenuate bacterial growth (Supplementary Fig. 11). In a systemic challenge, the HphR mutant was avirulent whereas mice infected with an HphA mutant exhibited a delay in onset of symptoms and mortality (Fig. 6a–d). A pulmonary infection model was also used to test bacterial fitness since intranasal inoculation represents a more natural route of *A. baumannii* infection. The HphR mutant exhibited reduced bacterial lung burden, and was virtually undetectable in the spleen and blood suggesting the mutant is unable to disseminate and/or survive in the blood (Fig. 6e–g). The HemTR, HphA, and HsmA mutants showed a milder phenotype in terms of bacterial lung burden and systemic spread (Fig. 6e–g). While it is important to be cognizant of the polar effects in these

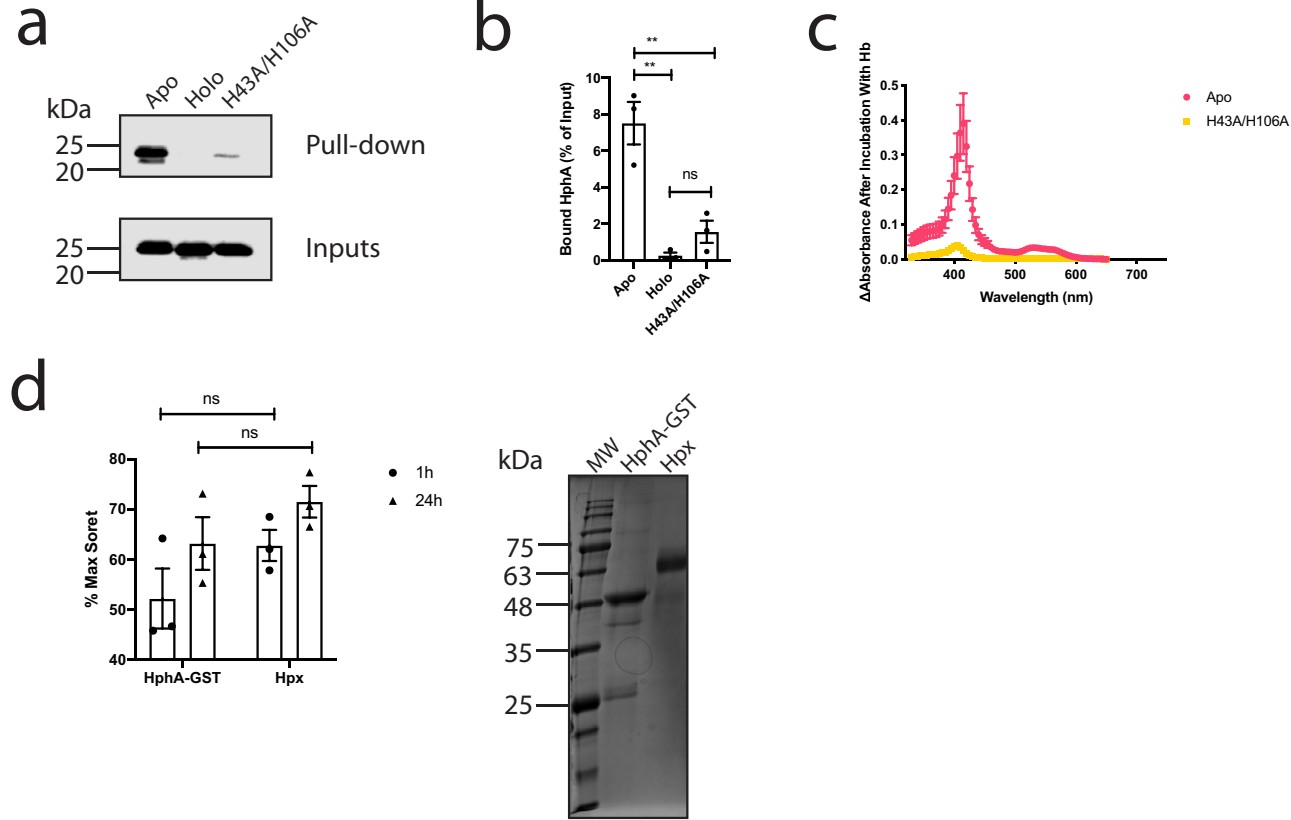

**Fig. 3 HphA interacts with hemoglobin and passively acquires heme. a** HphA pull-down showing binding of HphA to hemoglobin (Hb) resin. A representative anti-His Western from three distinct experiments is shown. Inputs represent a tenth of the protein used in the assay. **b** Quantification of average HphA band intensity normalized to input ± standard error of the mean from three independent pull-down experiments, a representative of which is shown in (**a**). Statistical significance determined by one-way ANOVA followed by Tukey's test. ns $P \geq 0.05$, **$P < 0.01$. Adjusted $P$ values are as follows: apo vs. holo $P = 0.0013$, apo vs. H43A/H106A $P = 0.0037$, holo vs H43A/H106A $P = 0.4949$. **c** The presence of heme bound HphA was detected by visible spectroscopy following incubation of apo-HphA with hemoglobin resin. The plotted curves represent the mean difference before and after incubation with beads ± standard error of the mean from three independent experiments. **d** GST tagged HphA and hemopexin (Hpx) were incubated with Hb beads to determine if heme stealing is passive. The change in Soret signatures, averaged over 413–415 nm and 414–415 nm for Hpx and HphA, respectively, were normalized to the protein concentration determined by Bradford assay and expressed as a percent of the max Soret signal determined by incubating an equal molar ratio of protein with hemin. Values plotted represent the average ± standard error of the mean from three different experiments. Note that the same max Soret signal and protein absorbance before incubation with Hb resin were used for normalization for all three experiments. Statistical significance determined by two-tailed unpaired student *t*-tests. Coomassie blue-stained SDS-PAGE gel of Hpx and purified HphA-GST (~50 kDa) is shown to the right.

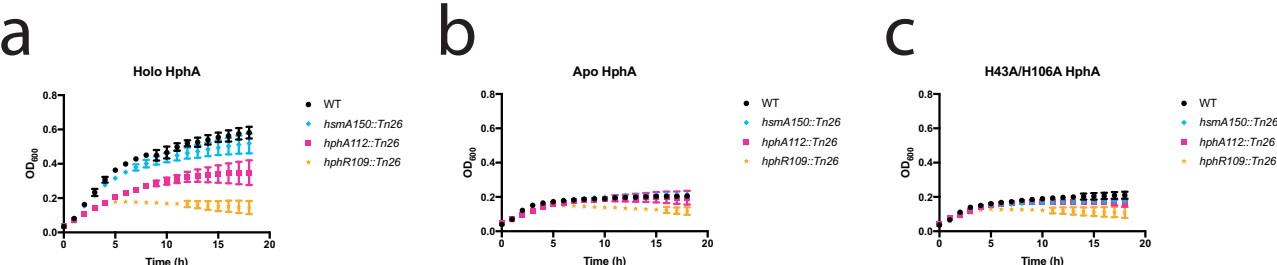

**Fig. 4 Purified holo HphA supports *A. baumannii* growth. a–c** 2.5 μM of purified holo (**a**), apo (**b**), or H43A/H106A (**c**) HphA was added to RPMI as the sole heme/iron source for growth of WT *A. baumannii* and transposon mutant strains. Values plotted represent the mean ± standard error of the mean from three experiments.

mutants and interpret these results within this context, a relative comparison of each strain suggests that HphR is essential for virulence while HphA and HsmA play supporting roles for full virulence.

## Discussion

Together these results show that heme cluster 2 functions as a high-affinity heme acquisition system. HphR functions analogously as a gateway for heme entry into the bacterial cell, enabling

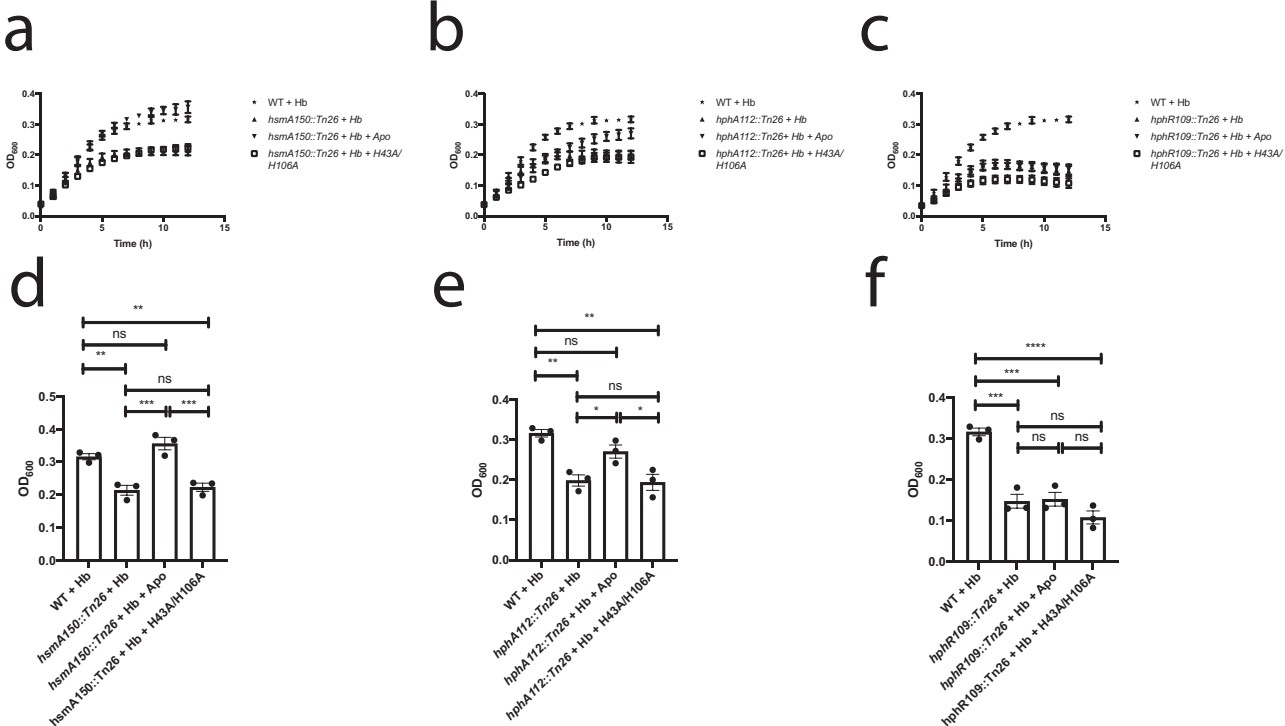

**Fig. 5 HphA can pirate heme from hemoglobin and support *A. baumannii* growth. a–c** To study the heme stealing properties of HphA, 2.5 µM purified WT and H43A/H106A HphA was added to RPMI media in combination with 50 nM hemoglobin (Hb). Growth of *hsmA150::Tn26* (**a**), *hphA112::Tn26* (**b**), and *hphR109::Tn26* (**c**) was measured at 37 °C. **d–f** $OD_{600}$ at the 12 h timepoint for *hsmA150::Tn26* (**d**), *hphA112::Tn26* (**e**), and *hphR109::Tn26* (**f**). Note: the same WT data was used in all panels for ease of comparison between WT and each individual mutant. All values plotted represent the mean ± standard error of the mean from three separate experiments, and results analyzed by one-way ANOVA followed by Tukey's test. ns $P \geq 0.05$; *$P < 0.05$; **$P < 0.01$; ***$P < 0.001$; ****$P < 0.0001$. Adjusted $P$ values are as follows: mutant + Hb vs. WT + Hb $P = 0.0043$ (**d**), $P = 0.0029$ (**e**), $P = 0.0002$ (**f**); mutant + Hb + apo vs. WT + Hb $P = 0.2717$ (**d**), $P = 0.2331$ (**e**), $P = 0.0003$; mutant + Hb + H43A/H106A vs. WT + Hb $P = 0.0076$ (**d**), $P = 0.0022$ (**e**), $P < 0.0001$ (**f**); mutant + Hb + apo vs. mutant + Hb $P = 0.0005$ (**d**), $P = 0.0434$ (**e**), $P = 0.9954$ (**f**); mutant + Hb + H43A/H106A vs. mutant + Hb $P = 0.9654$ (**d**), $P = 0.9962$ (**e**), $P = 0.3137$ (**f**); mutant + Hb + H43A/H106A vs. mutant + Hb + apo $P = 0.0008$ (**d**), $P = 0.0322$ (**e**), $P = 0.2337$ (**f**).

growth on hemoprotein sources and is absolutely critical for in vivo virulence, bacterial dissemination, and growth in the blood. Heme cluster 2 also encodes a secreted heme-binding protein, HphA, that functions hypothetically with its receptor, HphR, in heme transport across the OM (see Fig. 7, a hypothetical model of how the HsmA-HphA system operates). Although not strictly required for virulence, HphA is an accessory factor that enhances the process of heme uptake.

HsmA is critical for the translocation of HphA across the OM, and this agrees with a recent study that shows that Slams represent a novel class of secretion system (Type XI) for soluble and lipoproteins[23]. Slams have been found to cluster into groups based on sequence similarity and this correlates with factors like environmental niche of the corresponding bacteria and type of substrate such as secreted versus lipidated[23]. The current body of knowledge on Slams comes from focusing on the ones that translocate surface lipoproteins, and our work adds to the diversity of Slam-dependent substrates. It remains to be determined if *A. baumannii* Slam translocates additional lipidated or secreted substrates, which perhaps accounts for its slightly reduced virulence compared to the HphA mutant.

While it is clear *A. baumannii* secretes HphA, it is interesting that the HphA signal peptide contains a canonical lipobox motif. Signal peptide analysis software[24] predicts that the HphA signal peptide is processed by Signal Peptidase I thereby excluding processing by the lipoprotein specific Signal Peptidase II and trafficking by the downstream Lol pathway. These findings merit a complete examination of the HphA translocation pathway

including chaperones that may be involved in maintaining HphA in an inactive state until it is secreted.

Once secreted, HphA has the dual capacity of binding hemoglobin and scavenging heme. Based on the pull-down results, we propose that apo HphA binds to Hb largely through its heme-Fe coordination residues, H43 and H106. This mode of binding is plausible as the loop containing H43 is flexible and could wedge itself into the heme-binding site, positioning HphA to capture spontaneously released heme using H106 in a relatively quick step. The loop containing H43 then dissociates from Hb and clamps down on heme in a relatively slower binding step. This biphasic mode of heme binding has been proposed for the HasA hemophore based on stopped flow UV–Vis spectroscopy experiments[25]. It is not clear why HphA associates with Hb since HphA acts as heme scavenger in our experimental conditions, and could achieve this without Hb binding. One possibility is that Hb binding localizes HphA to the site heme release. Another explanation is that HphA blocks haptoglobin from binding to Hb. Haptoglobin sequesters Hb released from lysed red blood cells eliminating Hb from the blood, and is thought to suppress heme release by binding close to the heme pocket[26].

HphA belongs to a growing list of hemophores found in both Gram-positive and Gram-negative organisms. The IsdX1 hemophore secreted by the Gram-positive *Bacillus anthracis* contains the well-studied NEAr-iron transport (NEAT) domain, which is a predominantly all β, immunoglobulin-like fold[27,28]. IsdX1 coordinates heme through a single tyrosine. Despite the stark differences in structures and heme coordination chemistry, both IsdX1

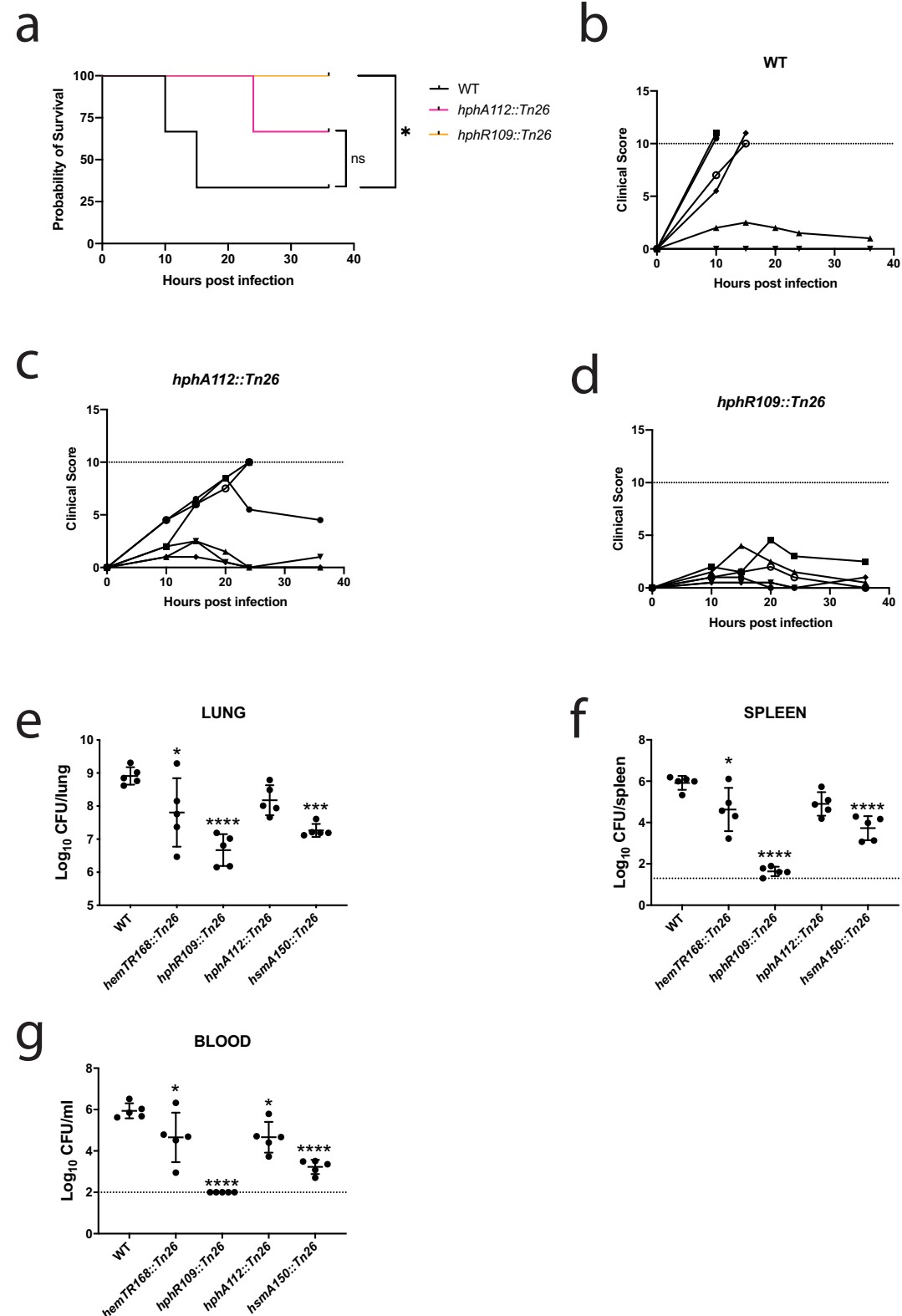

and HphA bind to heme and holo hemoglobin. A $3_{10}$ helix that sits atop of the heme in IsdX1 contains residues that are important for heme and hemoglobin binding[28]. This study demonstrated that these are connected processes that do not require mutually exclusive residues involved in either heme or Hb binding[28]. We have similarly shown that the heme coordination residues in HphA, H43 and H106, are also crucial for Hb binding.

A striking difference is that IsdX1 was shown to bind Hb with a micromolar $k_d$ determined by surface plasmon resonance, but was not captured by pull-down suggestive of a transient interaction[27]. On the other hand, the HphA-Hb interaction was detected by pull-down, but not by biophysical interaction tools.

Similar hemophores also exist in Gram-negative bacteria like the periodontal pathogen, *Porphyromonas gingivalis* which

**Fig. 6 HphR is essential for virulence, while HphA and HsmA are accessory factors enhancing infection. a** Survival of mice following intraperitoneal injection with ~2 × 10⁶ colony-forming units (CFU) of AB5075 (WT), or transposon mutants. Statistical significance determined by log-rank (Mantel-Cox) test. ns $P \geq 0.05$, *$P < 0.05$. WT vs. *hphA112::Tn26* $P = 0.1211$ and WT vs. *hphR109::Tn26* $P = 0.0191$. **b–d** Clinical scores of each mouse group after intraperitoneal infection with WT (**b**), *hphA112::Tn26* (**c**), or *hphR109::Tn26* (**d**). Dotted line indicates threshold for euthanasia. **e–g** Groups of 5 BALB/c mice were infected with ~5 × 10⁷ CFU of AB5075 (WT) or transposon mutants via intranasal inoculation. Bacterial burdens in the lung (**e**), spleen (**f**), and blood (**g**) were determined by quantitative bacteriology at 24 h after inoculation. The data are presented as mean ± standard deviation ($n = 5$) and represent one of two experiments with similar results. The detection limit (dotted lines) for bacterial burdens was 1.3 log₁₀ CFU/organ for the lung and spleen and 2.0 log₁₀ CFU/ml for blood. Differences in the bacterial burdens were assessed by one-way ANOVA followed by Dunnett's post hoc multiple comparisons test *$P < 0.05$, ***$P < 0.001$, or ****$P < 0.0001$ vs. WT. Adjusted $P$ values correspond to each of the following groups vs. WT: *hemTR::Tn26* $P = 0.0207$ (**e**), $P = 0.0130$ (**f**), $P = 0.0217$ (**g**); *hphR109::Tn26* $P < 0.0001$ (**e**), $P < 0.0001$ (**f**), $P < 0.0001$ (**g**); *hphA112::Tn26* $P = 0.1639$ (**e**), $P = 0.0561$ (**f**), $P = 0.0224$ (**g**); *hsmA150::Tn26* $P = 0.0007$ (**e**), $P < 0.0001$ (**f**), $P < 0.0001$ (**g**).

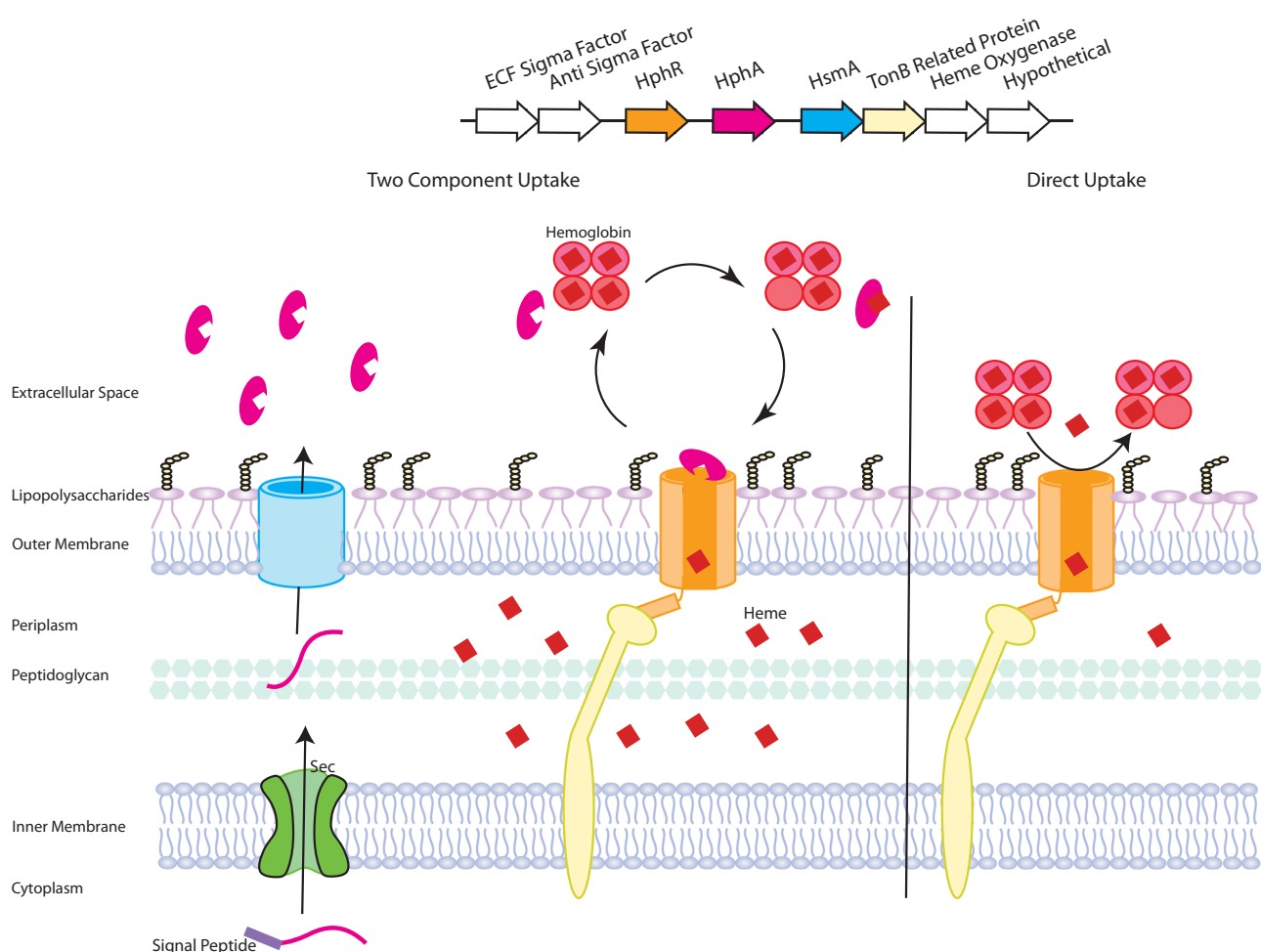

**Fig. 7 Working model of heme acquisition via heme cluster 2.** The model begins with the trafficking of HphA (magenta) across the inner membrane and into the periplasm via the sec translocon (green). Slam (blue) mediates the secretion of HphA into the extracellular environment. As we have shown, *A. baumannii* can tap into hemoglobin and human serum albumin as heme sources. HphA delivers heme to its receptor, HphR (orange), facilitating heme transport across the outer membrane, which is achieved with the concerted action of an inner membrane TonB complex (yellow). HphR still retains the ability to acquire heme from hemoprotein reservoirs, however, HphA likely enhances the efficiency of this process.

produces two hemophores as it lacks the biosynthetic machinery to produce heme. HmuY is an all β sheet protein lacking barrel structure that also binds to heme through bis-His coordination[29]. Although HmuY is exported and anchored to the cell surface, downstream proteolysis results in release into the extracellular space, functioning analogously to HphA. HmuY is a high-affinity heme-binding protein with a subnanomolar $k_d$, and can acquire heme from hemoglobin and albumin[29–31]. *P. ginigivalis* secretes proteases greatly enhancing release of heme from hemoglobin stores and capture by HmuY, which is resistant to degradation[29,30]. HusA is the second hemophore secreted by

*P. ginigivalis* and is a completely helical protein that binds heme with a micromolar $k_d$[32,33]. Current evidence suggests a unique mode of heme binding compared to HphA and HmuY in which HusA does not coordinate iron, and instead binds to porphyrin[33].

While structurally distinct, HphA is also similar to the HasA hemophore secreted by *Serratia marcescens* with analogous systems in other Gram-negative pathogens[14]. HasA is known to acquire heme from hemoglobin, and this process is thought to be passive given the hemophore's high affinity for heme ($k_d = 18$ pM) rather than direct extraction[34]. Support for the former hypothesis comes from a study of the heme transfer rate between

metHb and apo HasA, which was similar to the rate of spontaneous heme release from metHb[25]. In agreement with the IsdX1 and HphA findings, choosing the appropriate method is critical for detecting an interaction between HasA and Hb. Analytical ultracentrifugation did not prove useful in detecting an interaction between the two hemoproteins, while NMR demonstrated that holo HasA interacts with metHb through 2 distinct putative regions located on the surface and the loops containing the heme coordination residues, H32 and Y75[35,36]. Heme transfer between apo HasA and metHb prevented study of the complex by NMR[36]. Comparison of the apo and holo HasA structures revealed that the loop containing H32 is highly flexible similar to the HphA loop that coordinates heme through H43[36]. The H32 loop can swing out of the HasA heme-binding pocket ~20 Å forming a contiguous surface with the other putative Hb interaction region[36,37]. It is not known, however, if H32 is crucial for this interaction or serves as an acceptor site for heme transfer from Hb[36].

It was recently published that the human commensal *Haemophilus haemolyticus* produces a hemophore named hemophilin with 39% sequence identity to HphA[15]. Hemophilin adopts a similar structure as HphA consisting of an N-terminal heme-binding clamp and C-terminal barrel. Differences between heme coordination modes exists between the two proteins. HphA binds to heme through two His residues whereas hemophilin coordinates heme through a single His, and the other axial position contains a chloride ion and Arg that forms cation–π interactions with heme. While this study has not directly shown that *H. haemolyticus* secretes hemophilin via a Slam protein, another recent paper independently identified a putative Slam in *H. haemolyticus* and confirmed that an HphA homolog in *Xenorhabdus nematophila* is secreted by Slam[23]. Interestingly, *H. haemolyticus* can outcompete pathogenic *Haemophilus* species that do not produce hemophilin[15,38]. Like *Haemophilus*, *A baumannii* colonizes the respiratory tract, which is considered a heme scarce environment[17]. However, Hb, Hpx, and albumin have been detected in human nasal secretions[39] and therefore, it is plausible that *A. baumannii* strains that possess the Slam gene cluster may have a better chance of competing against *H. haemolyticus* for host heme in this shared niche.

As mentioned previously, HphA is also highly structurally similar to the other known class of Slam-dependent substrates, SLPs for which structures have been solved from *Neisserial* species. SLPs have diverse functions including heme uptake. Hemoglobin-Haptoglobin Utilization Protein A (HpuA) is an SLP that works cooperatively with its cognate TBDR (HpuB) to import heme from hemoglobin[40,41]. HpuA is required for high-level binding of HpuB to hemoglobin and subsequent release of the substrate[41]. Although HpuA and HphA both bind Hb, HpuA has the added ability to bind heme[42]. While HpuA does not extract heme from Hb, HpuA likely facilitates uptake by increasing the local concentration of hemoglobin close to the surface for extraction by HpuB.

In the last stage of our model, HphA delivers heme to its OM receptor HphR, and this is a plausible mechanism based on the X-ray crystal structure of the HasA hemophore bound to its receptor, HasR[43]. Since HasR has a much lower affinity for heme compared to HasA, HasR uses its extracellular loop to dislodge heme from the HasA heme-binding pocket[43,44]. HasR binding to HasA also induces conformational changes in HasA and disrupts heme coordination[43]. Being involved in high-affinity heme uptake, HphA likely interacts with HphR directly to induce heme release and permit efficient heme transport across the OM. Our working model also takes into account that HphR is also able to function independently in heme piracy. Likewise, HasR functions on its own in heme uptake from Hb while HasA enhances the

efficiency of heme acquisition by supporting growth on 100-fold lower Hb concentration[45].

An overall conclusion from this study is that heme uptake is critical for *A. baumannii* dissemination and/or survival in the blood. This is in agreement with a previous study that showed that the hypervirulent and septicemia causative strain LAC-4 has a growth advantage in serum compared to ATCC 17978, which is considered a non-bacteremic strain in immunocompetent mice[8]. This is attributed to the enhanced ability of LAC-4 to acquire heme as a result of its *hemO* gene cluster. Heme utilization appears to be the key to the successful spread, as both LAC-4 and 17978 are similar in their ability to produce iron-chelating siderophores, and steal iron from the serum glycoprotein, transferrin[8,9,46].

While heme is important for *A. baumannii* physiology and virulence, it is important to recognize that this pathogen is a generalist and has an arsenal of methods for acquiring diverse forms of iron during infection. *A. baumannii* encodes three main gene clusters that produce up to 10 different siderophores for chelating ferric iron[46]. The major and most well-studied siderophore acinetobactin was shown to be important for intracellular survival in epithelial cells and required for virulence in a mouse sepsis model[47]. A recent systematic study looked at the contributions of each cluster through individual knockout mutants[46]. They discovered redundancy among the gene clusters, but only acinetobactin is required for optimal growth in iron-restricted media and survival in a mouse sepsis model[46]. Analysis of gene expression in different organs revealed that all three clusters are upregulated in this infection model[46]. Multiple siderophores are thought to be important for adapting to different host niches, and while this may be true the authors acknowledge analyzing gene expression in whole tissues is not sensitive enough to detect subtle changes in siderophore levels[46]. Furthermore, *A. baumannii* encodes an Feo system dedicated to ferrous iron uptake. $Fe^{2+}$ is found in low oxygen and acidic environments and the Feo system may help *A. baumannii* thrive in these niches[48]. FeoA is a cytoplasmic protein of unknown function and was discovered to be overexpressed in a mouse lung infection model and its deletion attenuated virulence[48,49]. FeoB is the ferrous iron inner membrane importer and was shown to be required for growth in heat-inactivated human sera, however, was not required for virulence in a mouse sepsis model[48].

Although *A. baumannii* can acquire iron from diverse sources, heme plays a central role in virulence. It is not clear why *A. baumannii* possesses two heme uptake clusters, but examining heme acquisition systems in other organisms provides some perspective. *Serratia marcescens* encodes two heme receptors, HemR and HasR. The latter hemophore dependent TBDR supports growth on lower Hb concentrations compared to HemR, which is a lower affinity receptor system[50]. This enables *S. marcescens* to respond to a wider range of heme concentrations that may be encountered in the host[50]. *P. aeruginosa* also encodes two heme uptake systems and it has been proposed that they fulfill different roles based on experiments indirectly measuring labeled heme uptake in heme receptor mutants[51]. The non-hemophore-dependent receptor PhuR is considered the dominant heme receptor that brings in the bulk of heme into the cell while the HasA hemophore and its receptor HasR act as a sensitive heme sensor regulating expression of genes involved in heme acquisition[51]. Multiple heme acquisition systems in *A. baumannii* may enable it to use a broader range of host heme concentrations, and fine-tune gene expression. Heme cluster 2 would enable *A. baumannii* to acquire heme when the pathogen first invades the bloodstream. Heme is likely to be extremely limiting during initial colonization of the nasal mucosa or entry into blood where most hemoglobin is found inside intact red

blood cells. Under these conditions, heme cluster 2 would facilitate initial heme scavenging supporting dissemination and also prime the cell by promoting expression of heme uptake genes prior to the secretion of hemolysins that would enhance its access to hemoglobin stores and result in rapid growth. With an abundance of heme sources, we suspect that *A. baumannii* can capitalize on the lower affinity receptor, HemTR, to facilitate additional heme uptake.

HemTR is involved in heme uptake demonstrated by redundancy with HphR at high Hb levels. However, it is also possible that heme is not the primary substrate for heme cluster 1 and that the encoding machinery can be co-opted for heme transport. For example, *E. coli* contains periplasmic binding proteins and an inner membrane permease that transports both peptides and heme[19]. The finding that heme oxygenase from cluster 2 is required for the breakdown of heme and access to iron also agrees with the idea that heme cluster 1 does not primarily function to import heme[14]. However, the intact heme molecule may be used directly by the bacterium such as incorporation into newly synthesized proteins as opposed to being metabolized.

Clearly the Slam containing heme acquisition operon makes *A. baumannii* more virulent in mice infection models, and is prevalent in hospital settings, with 60% of clinical strains possessing the *hemO* cluster[14]. Investigating the mechanisms of heme uptake, an understudied area of *A. baumannii* biology, can be exploited in the future for applied strategies to prevent or treat *A. baumannii* infections via specific antibiotics or vaccines.

## Methods

**Bacterial strains, plasmids, and antibodies**. See Supplementary Table 2 for a summary of bacterial strains, plasmids, and antibodies used in this study.

**Plasmid construction**. All primers used in this study are listed in Supplementary Table 3. For HphA surface expression experiments, HphA and HsmA were cloned into pET52b and pET26b vectors, respectively, by restriction-free cloning, which uses primers with 5′ ends complementary to the vector and 3′ ends specific for the gene of interest[52]. Both vectors contained a pelB signal peptide replacing both endogenous signal peptides. HsmA was cloned with a 6X His tag, and a C-terminal Flag tag was inserted in the HphA construct by round the horn site-directed mutagenesis, which uses primers with 5′ ends containing the insert[53]. Following PCR and gel extraction, the ends are phosphorylated with T4 polynucleotide kinase (NEB) and joined with T4 DNA ligase (NEB) while the parental template is digested with DpnI (Thermo).

For purification, crystallization and functional studies, HphA was cloned without its signal peptide (the first 21 amino acids), into a pET28a vector with an N-terminal 6X His tag. Amino acid substitutions were introduced by QuikChange site-directed mutagenesis (Agilent). HphA was also restriction free cloned into a homemade pET52b vector containing a C-terminal PreScission protease cleavage site followed by a GST tag.

For the *E. coli* growth assay, the stretch of DNA encoding HphR, HphA, HsmA, and TonB was amplified from *A. baumannii* LAC4[9] genomic DNA and cloned into pHERD20T by exponential megapriming (EMP) PCR cloning[54]. While similar to RF cloning, EMP cloning relies on one primer that has a complementary region to the vector and can clone larger inserts. Two rounds of EMP cloning were used to first insert HphR into pHERD20T, followed by HphA, HsmA and TonB. HphR contains a cryptic start codon, and three additional steps of round the horn site-directed mutagenesis were required to insert its endogenous signal peptide followed by a 6X His tag. A HphR deletion construct containing HphA, HsmA, and TonB was made by inverse PCR[55].

Complementation of *A. baumannii* transposon mutants was achieved by cloning the genes encoding HphA, HsmA, and HphR into the pVRL2Z vector designed for expression in multidrug resistant clinical strains[56]. Cloning was done using the restriction enzymes XhoI (Thermo) and NotI (NEB).

**Preparation of hemoproteins for growth assays**. Human hemoglobin (Sigma) was dissolved in PBS and fractionated via a Superdex 75 column (GE) to remove impurities such as free hemin. Concentration of hemoglobin determined using $\varepsilon_{406} = 167$ mM$^{-1}$ cm$^{-1}$ for methemoglobin[57]. Human serum albumin (Sigma) was dissolved in PBS and treated with 5% chelex for 2 h at 4 °C prior to use, and filter sterilized. Hemin from bovine (Sigma) was prepared in 0.2 M NaOH and treated with 5% chelex for 2 h at 4 °C prior to use, and filter sterilized. Human serum albumin complexed to hemin was prepared by mixing the two at a molar ratio of 2:1.

**A. baumannii growth assay**. *A. baumannii* strain AB5075 and single transposon knockout mutants were obtained from the University of Washington[16,58]. Double transposon mutants were constructed by Cre recombinase-mediated excision of the antibiotic marker leaving behind a scar, followed by transformation with genomic DNA from a second mutant[16,59]. pABcre plasmid encoding Cre recombinase was introduced into *A. baumannii* HemTR mutant by electroporation and transformants were sequentially selected on LB with 25 μg/mL rifampicin and 5 μg/mL tetracycline for presence of pABcre and loss of the transposon, respectively. Tetracycline sensitive transformants were cured of pABcre by growing to stationary phase in the absence of rifampicin. Genomic DNA from the HphR mutant was introduced into the cured HemTR mutant by motility based natural transformation[59,60]. This involved mixing 2 μg genomic DNA with bacteria in PBS, inoculating the suspension into wells punctured into solid media containing 0.5% agarose, 0.5% tryptone, and 0.25% NaCl. Following incubation at 37 °C for 18 h, transformants were selected on LB with 5 μg/mL tetracycline, and verified by colony PCR using a transposon-specific primer (T26) and two gene-specific primers (Supplementary Table 3).

Strains were iron starved overnight on LB plates supplemented with 200 μM 2,2′-dipyridyl at 37 °C. Bacteria were resuspended in RPMI-1640 (Sigma) containing human hemoglobin or human serum albumin alone or in complex with hemin. Growth was monitored in a 96-well plate using a VICTOR Nivo (Perkin Elmer) or Cytation 5 (BioTek) plate reader at 37 °C.

Transposon mutants were complemented with empty pVRL2Z or vector containing genes encoding HsmA, HphA, or HphR via electroporation using a method adapted from Lucidi et al.[56]. Briefly, *A. baumannii* strains were grown at 37 °C for 24 h, harvested, and washed twice with 10% glycerol. Electroporation was performed with a BioRad GenePulser using the 2.5 kV, 200 Ω, 25 μF setting, and 0.2 cm cuvettes (BioRad). Cells were immediately recovered in 1 mL SOC medium for 1 h at 37 °C. Transformants were selected on low salt LB agar with 500 μg/mL zeocin (Thermo). To assess heme-dependent growth, complemented strains were grown in low salt LB supplemented with 500 μg/mL zeocin. At OD$_{600}$ of 0.6, L-arabinose and 2,2′-dipyridyl were added to 2% and 200 μM, respectively, and growth continued at 37 °C for an additional 18–20 h. Cells were then harvested and resuspended in RPMI supplemented with 500 μg/mL zeocin, 2% L-arabinose and either 3 μM holo HphA or 75 nM human hemoglobin. Cells were also resuspended in tryptic soy broth with 500 μg/mL zeocin and 2% L-arabinose as an iron-rich control. Growth was monitored in a 96-well plate using the VICTOR Nivo plate reader (Perkin Elmer) or Cytation 5 (BioTek) plate reader at 37 °C.

**E. coli growth assay**. *E. coli* enterobactin knockout (entF724—Keio collection[61]) transformed with empty pHERD, vector containing HphR-HphA-HsmA-TonB (abbreviated RAAB) or pHERD containing HphA-HsmA-TonB (abbreviated AAB) were grown overnight in LB supplemented with kanamycin and ampicillin. Cells were harvested and washed with M63 media and inoculated 1:100 to M63 supplemented with 0.4% glycerol and 1 mM MgSO$_4$. Following 4–5 h of growth at 37 °C (OD$_{600}$ ~ 0.2), L-arabinose was added to a final concentration of 0.002% and growth was continued at 37 °C for an additional 21 h. Cells were then inoculated into M63 media containing hemoglobin or YT media. Growth was monitored in a 96-well plate using the VICTOR Nivo plate reader (Perkin Elmer) at 37 °C.

**E. coli surface lipoprotein translocation assay**. *E. coli* strain C43(DE3)[62] cells were co-transformed with pET26 N-terminal His-tagged HsmA and pET52 C-terminal FLAG-tagged HphA or empty pET26 and pET52 C-terminal FLAG-tagged HphA. Cells were grown overnight at 37 °C in auto-induction media[63] supplemented with ampicillin and kanamycin. Cells were harvested by centrifugation at 1500 × g, and washed with PBS containing 0.9 mM MgCl$_2$. Cells were incubated with α-FLAG (1:200 dilution from rabbit serum; Thermo) for 1 h at room temperature, followed by washing in PBS + 0.9 mM MgCl$_2$. Cells were incubated with R-PE conjugated anti-rabbit IgG (Rockland) diluted 1:200 in PBS + 0.9 mM MgCl$_2$ for 1 h at room temperature, followed by washing with PBS + 0.9 mM MgCl$_2$ to remove excess antibodies. Cells were resuspended in 200 μL PBS + 0.9 mM MgCl$_2$, and fluorescence measurements recorded using a BioTek Synergy plate reader with a 488 nm excitation and 575 nm emission.

**A. baumannii secretion assay**. Empty pVRL2Z or vector containing *hsmA* was introduced into WT and *hsmA150::Tn26 A. baumannii* strains by electroporation using the above method.

Transformants were diluted 1:100 in low salt LB with 500 μg/mL zeocin and grown to OD$_{600}$ 0.6. L-arabinose and 2,2′-dipyridyl were added to a final concentration of 2% and 200 μM, respectively, and growth continued for 18–20 h at 37 °C. Cells were collected by centrifugation at 1500 × g for 10 min. Cells were resuspended in the same volume of water and added to SDS loading buffer, and the supernatant was passed through a 0.22 μm syringe filter (Millipore) prior to mixing with SDS loading buffer. Samples were analyzed by SDS-PAGE and transferred to PVDF membrane. Blots were probed with 1:10,000 goat α GroEL or 1:2500 mouse α HphA, followed by 1:5000 peroxidase linked α goat or mouse antibodies. Raw western blot images for all relevant experiments are provided in Supplementary Fig. 13.

**Purification of HphA**. *E. coli* strain BL21(DE3) was transformed with pET28 N-terminal 6X His tagged HphA lacking its signal peptide, and grown overnight in LB with kanamycin at 37 °C. Cells were inoculated into 2YT media and grown to an OD of 0.8 prior to induction with 0.5 mM IPTG at 20 °C for 18 h. Cells were harvested and lysed via sonication in 25 mM Hepes pH 7.5, 200 mM NaCl with lysozyme, DNase I, 1 mM benzamidine, and 2 mM PMSF. Lysate was centrifuged at 39,810 × *g* for 1 h to remove debris, and filtered with 0.45 μm syringe filter (Millipore) prior to batch binding overnight at 4 °C with Nickel-NTA resin (Thermo). Beads were washed with 20 mM imidazole prior to elution with 250 mM imidazole. HphA was concentrated using a 25 kDa (Viva Spin) molecular weight cutoff, and injected onto a Superdex 75 (GE) column equilibrated with 20 mM Hepes pH 7.5, 100 mM NaCl. HphA was then concentrated again prior to crystallization.

SHuffle T7 competent *E. coli* was transformed with pET52 HphA-GST and grown overnight in LB with ampicillin at 37 °C. SHuffle cells were grown in 2YT media to an OD of 0.5 followed by addition of 0.5 mM IPTG and continued growth at 20 °C for 18 h. Cells were harvested and sonicated in 25 mM Hepes pH 7.5, 200 mM NaCl supplemented with lysozyme, DNase I, 1 mM benzamidine, and 2 mM PMSF. Lysate was clarified by centrifugation at 31,000 × *g* for 1 h, and incubated with glutathione sepharose 4 fast flow resin (GE) overnight at 4 °C. The resin was washed with resuspension buffer before elution with fresh 10 mM reduced L-glutathione prepared in resuspension buffer and adjusted to pH 8.

**Stripping holo-HphA of heme**. The protocol for removing heme is based on Fanelli et al. and Espinas et al.[21,22] Briefly, Nickel-NTA purified HphA was treated with an ice-cold solution of 80% acetone and 20% 2 M HCl. Precipitated HphA was sequentially dialyzed against distilled water, 1.5 mM NaHCO₃, and 20 mM Hepes pH 7.5, 100 mM NaCl prior to injection onto a S75 gel filtration column equilibrated with the latter Hepes/NaCl buffer. HphA was concentrated using a 25 kDa (Viva Spin) molecular weight cutoff.

Similarly, apo HphA-GST was prepared by acid-acetone treatment followed by fractionation on a S75 gel filtration column. However, holo HphA-GST was added to an ice-cold solution of ~100% ice-cold acetone containing a small volume of 12 M HCl (final HCl concentration is 30 mM)[64].

**Crystallization of holo and apo HphA**. Holo and apo His tagged HphA was initially screened using a Gryphon robotic drop setter (Art Robbins) against MCSG and JCSG + commercial screens. Holo crystals were generated by sitting drop vapor diffusion, in a 1:1 ratio of protein (9.7 mg/mL) to precipitant (100 mM sodium acetate pH 4.6, 3.5 M sodium formate), and further optimized by streak seeding using a cat whisker in wells containing 2:1 ratio of protein (25 mg/mL) to precipitant (100 mM sodium acetate pH 4.15, 3.25 M sodium formate). Apo crystals were obtained by sitting drop vapor diffusion, from initial screens with a 1:1 ratio of protein (13.9 mg/mL) and precipitant (P2₁2₁2₁ space group: 30% (w/v) PEG 8 K, 0.1 M sodium acetate pH 4.5, 0.2 M lithium sulfate; C222₁ space group: 100 mM Hepes pH 7, 10% (w/v) PEG 6 K). Crystals were cryoprotected by quicks soaks in reservoir supplemented with 20% glycerol and flash-frozen with liquid nitrogen.

**HphA data collection and structure determination**. Data collection on holo and apo HphA crystals frozen at 105 K was performed on 08ID-1 and 08B1-1 at the Canadian Light Source (CLS), respectively. Diffraction data from 360 images with 1° oscillations were collected at wavelengths of 0.9796 Å (holo) and 0.9789 Å (apo). Data were processed with XDS and truncated to a resolution of 1.53 Å (holo), 1.49 Å (apo-P2₁2₁2₁), and 1.87 Å (apo-C222₁). A model of holo HphA was obtained by Fe-SAD, using a 2.1 Å dataset collected at 1.7343 Å, Phenix Autosol and Autobuild. The structures of apo HphA were obtained by molecular replacement using holo HphA as the search model and Autobuild. Final structures were generated after several rounds of model building and refinement using COOT, Phenix Refine, including TLS refinement yielding a final $R_{work}/R_{free}$ of 0.15/0.17 for holo and apo- P2₁2₁2₁ and 0.17/0.21 for apo C222₁. The web server, STRIDE[65], was used to assign secondary structures to holo and apo HphA for visualization by PyMOL.

**Size exclusion chromatography and multi-angle light scattering (SEC-MALS)**. SEC-MALS analysis on holo and apo HphA was done using a Malvern Viscotek GPCmax system connected to a Superdex 200 increase 10/300 GL gel filtration column (GE Healthcare). The column was equilibrated with buffer containing 20 mM Hepes pH 7.5, 100 mM NaCl. The scattered light intensity of the column eluate was recorded using a VE 3580 RI and Malvern 270 Dual detector. Bovine serum albumin at a concentration of 1 mg/mL was used for detector normalization. Molecular weights were calculated from Zimm plots using a protein refractive index increment (dn/dc) of 0.185 mL/g using the OmniSEC 5.10 software (Malvern).

**Heme extraction from hemoglobin beads**. Agarose beads conjugated to bovine hemoglobin (Sigma) were washed extensively with 20 mM Hepes pH 7.5, 100 mM NaCl. Beads were incubated with 30 μM WT or H43A/H106A HphA for 3 h at 4 °C. Beads were separated from the supernatant via centrifugation (16,100 × *g* for

2 min). Absorbance scans of the supernatants were recorded using a BioTek Synergy plate reader.

Heme scavenging from Hb was determined by comparing HphA with the known heme scavenger, hemopexin (Hpx; Athens Research and Technology). 20 μM Hpx prepared in 11.9 mM phosphate pH 7.4, 137 mM NaCl and 2.7 mM KCl, and 20 μM apo HphA-GST were incubated with hemoglobin resin at 4 °C. Supernatant samples were taken 1 h and 24 h after incubation and used for absorbance measurements via NanoDrop 2000c (Thermo). Soret peak absorbances (413–415 nm for Hpx and 414–415 nm for HphA) were averaged and normalized with respect to protein concentration determined by Bradford assay (Bio-Rad) using bovine lactoferrin and bovine serum albumin as standards, respectively. The amount of heme scavenged was expressed as a percentage of the maximal Soret observed upon mixing 1:1 apo Hpx or HphA with hemin.

**Pull down with holo hemoglobin resin**. Fifty microliters of hemoglobin resin and sepharose 4B (Sigma) as the matrix support used as an empty control were equilibrated with 20 mM Hepes pH 7.5, 100 mM NaCl and 0.05% Tween 20. HphA proteins at 15 μM were incubated with Hb resin for 2 h at 4 °C. The beads were washed extensively with equilibration buffer, resuspended in Orange G SDS loading buffer, and analyzed by SDS-PAGE followed by western blotting. PVDF membranes were blocked with Odyssey or Intercept blocking buffer (Li-COR). Membranes were then probed with 1:5000 α His diluted in block + 0.1% Tween 20, washed 4 × 5 min with PBS + 0.1% Tween 20, and probed with 1:10,000 α-mouse conjugated to IRDye 800 CW diluted in block + 0.1% Tween 20 + 0.01% SDS. Following 4 × 5 min washes with PBS + 0.1% Tween 20, the membrane was imaged by near-infrared fluorescence with an Odyssey CLx instrument (Li-COR).

**Pull down with apo hemoglobin resin**. Hemoglobin beads (Sigma) were treated with a solution of ice-cold 80% acetone and 20% 2 M HCl to remove heme, and incubated sequentially for 24 h in distilled water, 2 mM sodium bicarbonate and 20 mM Hepes pH 7.5, 100 mM NaCl. Fifty microliters of apo hemoglobin resin was incubated with HphA in 20 mM Hepes pH 7.5, 100 mM NaCl, and 0.05% Tween 20 for 2 h at 4 °C. The beads were washed extensively with equilibration buffer, resuspended in SDS load mix, and analyzed by SDS-PAGE followed by western blotting and detection by chemiluminescence.

**Pull down with AbHph-GST-coated resin**. Thirty microliters of glutathione resin equilibrated in 20 mM Hepes pH 7.5, 100 mM NaCl, 0.05% Tween 20 was mixed with 15 μM apo HphA-GST or free GST and 0.5 mg/mL bovine holo hemoglobin. Reactions were incubated at 4 °C for 2 h. The beads were washed with equilibration buffer three times for 5 min each, and protein eluted with 15 mM reduced L-glutathione. Samples were resolved by SDS-PAGE and visualized by Coomassie blue staining. Full SDS-PAGE gels for all relevant experiments are provided in Supplementary Fig. 14.

**Mouse sepsis infection model**. Eight- to ten-week-old specific-pathogen-free, male C57BL/6 mice were purchased from Charles Rivers Laboratories (St. Constant, Quebec, Canada). Mice were co-housed with enrichment and given water and food ad libitum. Room conditions were controlled under a 12/12 h dark/light cycle and 20–26 °C temperature with a relative humidity of 35–70%. Groups of 6 mice were inoculated intraperitoneally with 2 × 10⁶ CFU/mouse of WT, *hphA112::Tn26* or *hphR109::Tn26 A. baumannii* AB5075 in accordance with University of Toronto Animal Ethics Review Committee under protocol 20011319. Bacterial inoculums were prepared by growing the strains in LB supplemented with 150 μM dipyridyl (DIP) at 37 °C to OD₆₀₀ of 0.8, freezing and then growing in LB with 225 μM dipyridyl for 4–5 h. Mouse survival and clinical scores were monitored following the bacterial challenge, and mice were humanely euthanized if they had a cumulative score of 10 or higher. Mice were scored on a scale of 0–2 based on factors such as body weight decrease, grooming, posture, appearance of eyes and nose, breathing, unprovoked behavior, provoked behavior, dehydration, and diarrhea.

**Mouse pulmonary infection model**. Six- to ten-week-old specific-pathogen-free, female BALB/c mice were purchased from Charles Rivers Laboratories (St. Constant, Quebec, Canada). The animals were maintained and used in accordance with the recommendations of the Canadian Council on Animal Care Guide to the Care and Use of Experimental Animals, and the experimental procedures were approved by the institutional animal care committee (AUP#2016.11, Human Health Therapeutics Research Center, National Research Council Canada, Ottawa). Mice were kept on wood shavings in individually ventilated cages (5 mice per cage) and given ultra-purified water and standard mouse food pellets ad libitum. Room conditions were controlled under a 12/12 h dark/light cycle and 20–26 °C temperature with a relative humidity of 40–60%.

For intranasal (i.n.) inoculation, freshly grown inocula were prepared for each experiment from frozen stocks of *A. baumannii* strains as adapted from Harris et al. and described here[66]. Briefly, the bacteria were grown overnight on cystine heart agar (CHA) plates, then a portion was transferred into tryptic soy broth (TSB) supplemented with 35 mg/L of 4,4′-dipyridyl (DIP, Sigma) and incubated at 37 °C for 2–3 h until an OD₆₀₀ of 0.85 was reached (mid-log phase). Bacterial cells were

then centrifuged and resuspended in 0.85% saline at the desired inoculation concentration, based on microscope cell counts.

Mice were lightly anesthetized with isoflurane inhalation and intranasally inoculated with $5 \times 10^7$ CFU of different strains of *A. baumannii* in 50 μL saline according to van Faassen et al.[67]. Actual inoculum concentrations were confirmed by plating 10-fold serial dilutions on Brain Heart Infusion (BHI) agar plates. Clinical signs of the mice were monitored. Mice were sacrificed at 24 h post-inoculation and the lungs, spleen, and blood were aseptically removed. Lung and spleen homogenized in 2 ml sterile saline using aerosol-proof homogenizers, and blood was directly diluted 10-fold in sterile distilled water. Aliquots (100 μl) of 10-fold serial dilutions of the homogenates or blood were cultured on BHI plates to quantify the number of viable *A. baumannii* in the respective organs[68].

**Statistics**. Statistical analyses were performed using Prism 9 (GraphPad Software). Growth assay experiments were analyzed by one-way analysis of variance (ANOVA) followed by Tukey's post hoc test. Differences in fluorescence intensity for HphA cell surface localization assay were determined by an unpaired student's *t*-test. Statistical significance of the sepsis mouse experiment was determined using a log-rank (Mantel-Cox) test. Data for the pulmonary challenge are presented as means ± SD for each group ($n = 5$). Differences in the bacterial burdens were assessed by ANOVA followed by Dunnett's post hoc multiple comparison tests, when appropriate. Differences were considered significant when $P < 0.05$.

**Reporting summary**. Further information on research design is available in the Nature Research Reporting Summary linked to this article.

## Data availability

Atomic coordinates and structure factors have been deposited in the Protein Data Bank (PDB) under accession codes PDB 7RED, PDB 7REA, and PDB 7RE4. Source data are provided with this paper.

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

## Acknowledgements

The authors thank H. Wyatt (University of Toronto), W. Houry (University of Toronto), D. Bona and K. Maxwell (University of Toronto), and C. Manoil (University of Washington) for reagents, antibodies, and strains. This work was supported by a CIHR grant awarded to TFM (PJT-148795). T.F.M. is supported by a Canada Research Chair in the Structural biology of Membrane Proteins. T.J.B. is supported by OGS. We also thank the Canadian Light Source staff at the Canadian Macromolecular Crystallography Facility for help with data collection using the 08ID-1 and 08B1-1 beamlines.

## Author contributions

T.J.B. and T.F.M. designed the experiments. T.J.B. performed experiments and analyzed the data. M.S. performed SEC-MALS, collected X-ray diffraction data, and processed the initial datasets. T.P.H. helped purify His tagged HphA proteins and perform the pull-down with holo hemoglobin beads. C.P. and HES helped purify HphA-GST protein and performed the reciprocal pull-down experiment with HphA-GST and hemoglobin. G.H., J.E.F., E.A.I., and S.K.A. conducted the mice challenge experiments, while S.D.G.O. and W.C. supervised the animal studies. The methods and statistical analysis used for the mouse pulmonary challenge were provided by G.H. and W.C. Y.H. performed the bioinformatics analysis leading to the discovery of a Slam in *A. baumannii*. T.J.B. wrote the manuscript with input from T.F.M.

## Competing interests

T.J.B., M.S., C.P., H.E.S., T.P.H., G.H., J.E.F., E.A.I., S.K.A., and W.C. declare no competing interests. T.F.M., S.D.G.O., and Y.H. are co-authors on a patent, "Slam polynucleotides and polypeptides and uses thereof".
