## [Peer Review File · Nature Communications]

A Slam-dependent hemophore contributes to heme acquisition in the bacterial pathogen *Acinetobacter baumannii*Reviewers' comments:

Reviewer #1 (Remarks to the Author):

This paper focuses on predicted heme uptake systems of *Acinetobacter baumannii* and their role in utilizing heme as an iron source for this pathogen. Initially, two gene clusters are listed as potential heme uptake systems, the second of which is demonstrated (Figure 1) to have a critical role in growth with a low concentration of hemoglobin (Hgb) as a sole iron source. An interesting observation from figure 1 is that the two TonB dependent transporters (TBDR) are redundant in abundant Hgb conditions. It would be relevant to the authors' goal to create the double TBDR mutant and explore its growth on abundant Hgb.

The bulk of the paper focuses on a lipoprotein encoded by the second gene cluster and the authors claim to demonstrate that (i) it is exported by another encoded protein (AbSlam); (ii) it binds heme, (iii) it binds Hgb and (iv) it can extract heme directly from Hgb.

The data supporting claim (i) is very clear, as would be expected from the group who discovered Slam. Claim (ii) is backed up by exceptional structural data combined with mutagenesis.

The binding of Hgb (claim (iii)) is supported by the pull downs, but there is an important missing control here. The use of the holo and double Histidine mutant make good negative controls and present interesting data as to the nature of the interaction with Hgb. However, false positives are possible in pulldown experiments where solubility is an issue (i.e. is apo AbSLP less soluble than the holo or mutant forms under these conditions?). Can the authors demonstrate that apo AbSLP is not 'pulled down' by empty agarose beads or by beads conjugated to another protein?

The stripping of heme from Hgb (claim (iv)) as opposed to high affinity scavenging of free heme is perhaps the hardest claim to prove. The binding of apo-, but not holo-ApSLP to Hgb supports this claim circumstantially and Figure 3c shows that heme can be taken from Hgb beads, but this does not distinguish between a scavenging vs pirating function. Hemopexin is a high affinity heme scavenging protein, not reported to take heme directly from Hgb. Can the experiment from Fig3c be repeated using hemopexin (or other heme scavenger) in combination with Hgb beads as a scavenging control? Measuring the affinity of ApSLP for heme and Hgb would also help illuminate its role. The Hgb interaction is presumably transient if it is indeed stripping heme, but it appears stable over the 2hr incubation at 4°C for the pull down experiment, so an affinity measurement could be possible. The affinity for heme should be straightforward to measure and would suggest whether ApSLP is able to compete for free heme.

The final section of the results importantly demonstrates the significance of this gene cluster and the TonB dependent transporter in the dissemination of *A. baumannii* from the lung in an infection model.

The results and discussion sections are very brief and some elements could be further explored and discussed:

- Fig4e-g: AbSlam deletion appears to have a more deleterious impact than AbSLP deletion. Is this significant? Are there other substrates of AbSlam that would explain it?
- A more detailed comparison with other bacterial hemophores should be included. HmuY is mentioned briefly, but not HasA. One of these has been suggested to extract heme from Hgb, the other is reported to rely on Hgb degradation and heme scavenging.
- The evidence of heme being important for bacterial dissemination in mice is interesting. Is anything known about other iron sources for *A. baumannii* and their requirement at different infection stages? Siderophores?
- The apo-ApSLP crystal structure contains three monomers: are these all identical? Or is does the displaced loop show flexibility?

Minor points:

- Figure 3: use the more standard Hb or Hgb abbreviation for haemoglobin
- Line 137: figures referenced should be 4e-g

- Results section: use subheadings for ease of navigation by readers

In conclusion, this is an interesting paper detailing some key features of a significant iron uptake system from an important human pathogen. To my mind, the main novelty is the unique nature of the SLP acting as a hemophore and the potential for this to strip heme from hemoglobin. This makes the system distinct from other bipartite iron uptake systems and hemophore-dependent systems. The mouse model results will also be of wider interest to host pathogen researchers. I recommend the paper is published after revision, providing the above points can be addressed.

Reviewer #2 (Remarks to the Author):

The manuscript examines a heme acquisition system in the opportunistic pathogen *Acinetobacter baumannii*. A single gene cluster is identified as responsible for heme-dependent growth enhancement *in vitro*. Integrated genetic, biochemical and structural analyses of a key element encoded in this cluster, AbSLP, support the model that the protein is a surface receptor binding hemoglobin and heme, facilitating the growth enhancement. Mutation of genes within the cluster affect virulence and fitness in murine infection models.

I am supportive of the manuscript but have some concerns with the genetic analysis. The study uses transposon mutants obtained from a published library to analyze the roles of heme acquisition genes in enhancing growth and virulence. There are issues with this approach, namely it requires ruling out the possibility that polar effects of the transposon mutation or unlinked mutations/alterations contribute to key phenotypes. Experiments examining effect of supplementing purified AbSLP in broth cultures mostly addressed these concerns with regard to *in vitro* growth phenotype, but this was not addressed in the animal model. This concern could be addressed by complementation experiments determining whether the *in vivo* phenotypes attributed to individual genes can be reversed by expressing a copy of the gene *in trans*.

Minor comments:

-the manuscript uses the Δ ("del") symbol to indicate the transposon mutants, but this symbol is usually reserved for true genetic deletions, not transposon insertions. The strains should be denoted by the name of the allele, or another accepted indicator of Tn insertion.

-line 124: A polar effect typically involves reductions in a co-transcribed gene that is promoter-distal to the mutation. As diagrammed, however, the gene encoding TBDR2 is upstream of the gene encoding AbSLP, so polarity of AbSLP mutation on TBDR2 expression (as suggested in the text) appears to be an unlikely explanation. Polar effects of mutations on promoter-distal genes is a more likely scenario and could be ruled out as described above.

-Was the same exact WT data presented in each panel of Fig. 3d-i? If so this should be clearly indicated.

-In the *in vitro* growth experiments, the data presented with linear y-axis show defects in overall growth yield related to heme acquisition mutants, but defects in logarithmic growth rate/doubling time are difficult to ascertain. Were these observed?

Reviewer #3 (Remarks to the Author):

The manuscript by Bateman et al. investigates the means by which *Acinetobacter baumannii*, an emerging opportunistic, and frequently multi-drug resistant pathogen, utilizes heme as an iron source. A previous bioinformatic analysis (Antunes LC, Imperi F, Towner KJ, Visca P. *Res Microbiol.*

2011 Apr;162(3)) revealed that some *A. baumannii* isolates possess two putative heme uptake clusters, heme cluster 1 and heme cluster 2 (hemO cluster), and Bateman and colleagues work towards elucidating the individual contributions of these two systems to heme-iron utilization. Through use of mutants disrupted for key components of each system, the authors find that only heme uptake cluster 2 contributes to the iron-dependent growth of *A. baumannii* on heme and hemoproteins. In beginning to characterize the machinery involved in heme uptake through cluster 2, the authors focus on two hypothetical proteins, a proposed surface-anchored lipoprotein (abSLP) and an associated surface protein assembly modulator (AbSlam). Both abSLP and AbSlam are required for growth on heme/hemoproteins as a sole iron source, and heterologous expression of these two proteins, along with TonB and a TonB-dependent transporter (TBDT2), is sufficient to facilitate heme utilization by *Escherichia coli*. In *E. coli*, abSLP-FLAG is found to be cell surface associated through labelling with an α -FLAG conjugated fluorophore and subsequent fluorescence detection. Following purification, the crystal structure of AbSLP was solved and essential residues in heme binding (H30 and H93) were identified. Lastly, the authors show that the TBDTs of both heme uptake systems are required for maximal survival and virulence of the bacteria *in vivo*, and that AbSlam, and to a lesser extent, AbSLP also contribute to the pathogenesis of *A. baumannii*. Similar to iron acquisition systems in *Neisseria*, the authors propose a model where abSlam facilitates surface presentation of AbSLP, which in turn increases the efficiency of heme extraction and capture. Following capture, heme is then imported by TBDT2 and TonB for utilization by the bacteria.

Overall, the manuscript is well-written, flows logically, and is pleasant to read. Given the recent interest by both the WHO and CDC in better understanding the basic biology of *A. baumannii*, the paper is topical. However, concerns about novelty are raised by the fact that similar findings were recently published by another group (Giardina BJ, Shahzad S, Huang W, Wilks A. 2019. Arch Biochem Biophys. Sep 15;672:108066), identifying AbSLP as a hemophore that facilitates heme acquisition by the bacterium. At the very least, the authors need to reframe their findings in the context of the preexisting literature. Additional questions, concerns, and suggestions to help improve the paper are below.

Major criticisms/corrections:

1. As noted above, the role of heme uptake clusters 1 and 2 in heme acquisition by *Acinetobacter baumannii* was recently addressed in a manuscript by the Wilks Lab, where heme uptake cluster 2 (referred to as the hemO cluster) was found to be required for the heme-dependent growth of *A. baumannii* whilst heme cluster 1 was found to be dispensable for this function. Figure 1 of the current manuscript effectively recapitulates figures 1 and 2 of the aforementioned manuscript, albeit with a cleaner experimental design. Further, Giardina et al perform hemin-agarose pulldowns on the iron-restricted spent culture supernatants of *A. baumannii* strains AB0057 and LAC-4, and identify homologues of AbSLP (ABLAC_16810/AB57_0988). The authors name these proteins AbHph for their heme-binding properties and propose, given the AbSLP-homologues were identified in the extracellular fraction that they function not as surface-anchored lipoproteins but rather as secreted hemophores. The authors need to cite Giardina et al. discuss how their findings fit into the context of what is already known about *A. baumannii* heme acquisition, and address the discrepancy in proposed function. The fact that apo-abSLP in combination with hemoglobin can restore growth of Slam and SLP mutants indicates that the bacteria is capable of utilizing abSLP as a hemophore. Expression of tagged abSLP in *A. baumannii* Δ abSLP followed by fractionation and Western Blotting would help identify the localization of abSLP in its native bacteria, instead of in a heterologous expression system.
2. Throughout the paper, common bacterial nomenclature is not adhered to in the naming of AbSLP, AbSLM, TBDR1, and TBDR2. Please rename so that the genes/proteins fit the convention of three lower case letters indicating the pathway/process and an uppercase letter specifying the gene/gene product. Given the AbSLP has already been named AbHph by the aforementioned group, it might be appropriate to incorporate part of this nomenclature in the new name.
3. An author on the current manuscript has previously published on the potential role of the hemO cluster on *A. baumannii* virulence, including bioinformatic analyses on the conservation of the hemO cluster in clinical isolates (Ou et al. (2015) Sci Rep 5(8643); de Léséleuc (2014) Int J Med Microbiol 304(3-4)). It is unclear why these papers were not cited as part of the background

literature/rationale for the study (the former appears to be cited only as a strain reference, and the latter does not appear to be addressed at all).

4. The authors make mention of the domain architectures of the SLP and Slam proteins; it would be helpful to have an actual schematic detailing this information in the supplement.

5. It is evident that AbSLP binds heme, and the authors suggest that the protein is also capable of binding both holo and apo-hemoglobin and that the same residues (H30 and H93) participate in these interactions. Heme-binding blocks the interaction with hemoglobin, which suggests heme and hemoglobin are bound at the same location. How do the authors propose that apo-hemoglobin is bound, if heme is not participating in the coordination?

6. In preparing the inoculum for the mouse infections, it appears that the bacteria were iron-starved first. Firstly, this should be noted in the main text, as it is not a common procedure in preparing bacteria for murine models of bacterial infection. Secondly, was this a necessary measure to obtain the phenotypes observed? It is possible that the mutants are already severely attenuated in vitro before being introduced to the mouse? Do the strains grow comparably under these conditions?

Minor criticisms/corrections:

Title: Capitalization is irregular (e.g. Heme is capitalized by acquisition isn't)

Title: Italicize *Acinetobacter*

Page 2, line 3: Here *Acinetobacter baumannii* is referred to as a nosocomial pathogen, but in the first line it is mentioned that it causes both hospital and community acquired infections. Perhaps for consistency "opportunistic pathogen" would be more appropriate in this context.

Page 2, line 12. Dissemination into the vasculature is not assessed directly. The lack of Δ TBDR2 bacteria in the blood could be due to survival and/or dissemination defects.

Page 2, line 18: Delete "disease in".

Page 2, line 20: Define the acronym "UTIs".

Page 3, line 28: This sentence is a bit confusing. Perhaps rephrase to something suggesting that *A. baumannii*'s demand for iron could potentially be exploited as a weakness.

Page 3, line 43: What kind of closer inspection? Presumably these proteins bear homology to neisserial SLP and Slam proteins and were identified through bioinformatics in the authors' earlier Nature Microbiology paper?

Page 3, line 44: How are they misannotated? What were they annotated as prior?

Page 5, line 72: Which TonB? Most *A. baumannii* strains express several.

Page 5, line 83: Delete "reconstitution assay."

Page 6, Line 92: Switch α to alpha, or beta to β for consistency.

Page 6, line 105: Add word "chromatography" and either capitalize all of the words or none of the words in Size Exclusion Chromatography with Multi-Angle Light Scattering.

Page 8, line 137: Change "Figure 4e-h" to "Figure 4e-g," as there is no panel "h".

Page 9, line 180: The authors discuss anchoring and subsequent cleavage of HmuY from the cell surface, where it then functions as a hemophore. As stated above, it is possible that the same is true of AbSLP.

Page 10, first paragraph: Given that heme cluster 1 lacks a heme oxygenase, unless one is encoded elsewhere in the genome, it is possible this cluster does not facilitate iron acquisition from heme (this is true of *A. baumannii* ATCC 17978).

Page 10, final paragraph: The authors suggest that the Slam containing heme acquisition operon (heme cluster 2, hemO cluster) makes *A. baumannii* more virulent in a mouse model and highlight the possibility that this cluster might be conserved in hospital isolates. Giardina et al found that ~60% of clinical isolates possess the hemO cluster.

Page 13, line 268 and page 14, line 277: Change "kDA" to "kDa"

Figure 1 caption: Add "(HSA)" after "human serum albumin".

Figure 1 caption: Add a space between "600" and "nm."

Figure 2 caption: Define "PE."

Figure 3 caption: Change "The plotted curves represents" to "represent"

Supplementary Figure 4: Shouldn't the MW of the holo protein be greater than the apo protein (see inset table)?

Supplementary Figure 4: Change kDA to kDa in y-axis label.

Responses to Reviewer Comments:

We thank the reviewers for their helpful comments and feedback. Reviewer comments are listed below in **BLACK** with responses to each comment in **RED** including line numbers or figures for relevant sections of the manuscript.

Here is a short-list of the additional reviewer-suggested experiments and data added to the previously submitted manuscript.

- Made a HphR/HemTR double mutant, and tested its growth in hemoglobin as the sole iron source (Figure 1c)
- Tested HphA's dependence on HsmA for secretion in *A. baumannii* (Figure 2a)
- Repeated the heme stealing experiment with hemopexin as a control, demonstrating passive heme stealing (Figure 3d)
- Reciprocal pull down using immobilized HphA-GST and hemoglobin in solution (Supplementary Figure 8b)
- Plasmid based rescue of *A. baumannii* transposon mutants (Supplementary Figure 11)
- Growth experiment showing iron starved *A. baumannii* is not attenuated prior to mice experiments (Supplementary Figure 13)
- Optimizing and repeating some of the supplementary *A. baumannii* growth experiments (Supplementary Figure 10, 12).

Reviewer #1 (Comments for the Author)

This paper focuses on predicted heme uptake systems of *Acinetobacter baumannii* and their role in utilizing heme as an iron source for this pathogen. Initially, two gene clusters are listed as potential heme uptake systems, the second of which is demonstrated (Figure 1) to have a critical role in growth with a low concentration of hemoglobin (Hgb) as a sole iron source. An interesting observation from figure 1 is that the two TonB dependent transporters (TBDR) are redundant in abundant Hgb conditions. It would be relevant to the authors' goal to create the double TBDR mutant and explore its growth on abundant Hgb. **We created the mutant and repeated the growth assay. See Figure 1c for details.**

The bulk of the paper focuses on a lipoprotein encoded by the second gene cluster and the authors claim to demonstrate that (i) it is exported by another encoded protein (AbSlam); (ii) it binds heme, (iii) it binds Hgb and (iv) it can extract heme directly from Hgb.

The data supporting claim (i) is very clear, as would be expected from the group who discovered Slam. Claim (ii) is backed up by exceptional structural data combined with mutagenesis.

The binding of Hgb (claim (iii)) is supported by the pull downs, but there is an important missing control here. The use of the holo and double Histidine mutant make good negative controls and present interesting data as to the nature of the interaction with Hgb. However, false positives are possible in pulldown experiments where solubility is an issue (i.e. is apo AbSLP less soluble than the holo or mutant forms under these conditions?). Can the authors

demonstrate that apo AbSLP is not 'pulled down' by empty agarose beads or by beads conjugated to another protein?

Solubility or nonspecific binding to empty beads are not issues as demonstrated in Supplementary Fig. 8a. We also performed the reciprocal experiment as demonstrated in Supplementary Fig. 8b.

The stripping of heme from Hgb (claim (iv)) as opposed to high affinity scavenging of free heme is perhaps the hardest claim to prove. The binding of apo-, but not holo-ApSLP to Hgb supports this claim circumstantially and Figure 3c shows that heme can be taken from Hgb beads, but this does not distinguish between a scavenging vs pirating function. Hemopexin is a high affinity heme scavenging protein, not reported to take heme directly from Hgb. Can the experiment from Fig3c be repeated using hemopexin (or other heme scavenger) in combination with Hgb beads as a scavenging control? Measuring the affinity of ApSLP for heme and Hgb would also help illuminate its role. The Hgb interaction is presumably transient if it is indeed stripping heme, but it appears stable over the 2hr incubation at 4°C for the pull down experiment, so an affinity measurement could be possible. The affinity for heme should be straightforward to measure and would suggest whether ApSLP is able to compete for free heme.

We repeated the heme extraction experiment using hemopexin as a control shown in Fig. 3d, and it revealed that AbSLP (now named HphA) steals heme passively. Unfortunately, we were unable to determine binding constant for heme and Hb due to issues with nonspecific binding.

The final section of the results importantly demonstrates the significance of this gene cluster and the TonB dependent transporter in the dissemination of *A. baumannii* from the lung in an infection model.

The results and discussion sections are very brief and some elements could be further explored and discussed:

- Fig4e-g: AbSlam deletion appears to have a more deleterious impact than AbSLP deletion. Is this significant? Are there other substrates of AbSlam that would explain it?

The difference between AbSlam (HsmA) and AbSLP (HphA) bacterial counts in the blood and spleen is statistically significant. Although beyond the scope of this study, it is certainly possible that HsmA translocates other substrates, and we acknowledge this is in the discussion section (lines 208-210).

- A more detailed comparison with other bacterial hemophores should be included. HmuY is mentioned briefly, but not HasA. One of these has been suggested to extract heme from Hgb, the other is reported to rely on Hgb degradation and heme scavenging.

We thank the reviewer for their comment, and have included a more thorough discussion of HphA in the context of what is known about other bacterial hemophores (lines 231-298).

- The evidence of heme being important for bacterial dissemination in mice is interesting. Is anything known about other iron sources for *A. baumannii* and their requirement at different infection stages? Siderophores?

We thank the reviewer for their comment, and have addressed the importance of different iron sources to *A. baumannii* virulence (lines 318-338). However, the contributions of these

other iron uptake systems at different stages of infection have not been defined, other than the fact that they are important for survival.

- The apo-AbSLP crystal structure contains three monomers: are these all identical? Or is does the displaced loop show flexibility?

The three monomers are essentially identical. We added Supplementary Fig. 6d to show the structural superposition of all the apo monomers.

Minor points:

- Figure 3: use the more standard Hb or Hgb abbreviation for haemoglobin. We thank the reviewer and have made the correction.

- Line 137: figures referenced should be 4e-g. We have made the correction.

- Results section: use subheadings for ease of navigation by readers. We thank the reviewer and have added subheadings.

In conclusion, this is an interesting paper detailing some key features of a significant iron uptake system from an important human pathogen. To my mind, the main novelty is the unique nature of the SLP acting as a hemophore and the potential for this to strip heme from hemoglobin. This makes the system distinct from other bipartite iron uptake systems and hemophore-dependent systems. The mouse model results will also be of wider interest to host pathogen researchers. I recommend the paper is published after revision, providing the above points can be addressed.

Reviewer #2 (Comments for the Author)

The manuscript examines a heme acquisition system in the opportunistic pathogen *Acinetobacter baumannii*. A single gene cluster is identified as responsible for heme-dependent growth enhancement *in vitro*. Integrated genetic, biochemical and structural analyses of a key element encoded in this cluster, AbSLP, support the model that the protein is a surface receptor binding hemoglobin and heme, facilitating the growth enhancement. Mutation of genes within the cluster affect virulence and fitness in murine infection models.

I am supportive of the manuscript but have some concerns with the genetic analysis. The study uses transposon mutants obtained from a published library to analyze the roles of heme acquisition genes in enhancing growth and virulence. There are issues with this approach, namely it requires ruling out the possibility that polar effects of the transposon mutation or unlinked mutations/alterations contribute to key phenotypes. Experiments examining effect of supplementing purified AbSLP in broth cultures mostly addressed these concerns with regard to *in vitro* growth phenotype, but this was not addressed in the animal model. This concern could be addressed by complementation experiments determining whether the *in vivo* phenotypes attributed to individual genes can be reversed by expressing a copy of the gene *in trans*.

We thank the reviewer for their comment, and have used our *in vitro* growth assays to show that plasmid based complementation can rescue most defects except for the AbSLP (HphA) mutant (Supplementary Figure 11). However, this mutant can grow when provided purified HphA (Supplementary Figure 10). We have not verified the rescued mutants in the *in vivo*

mouse model infection experiments due to economic and ethical constraints.

Minor comments:

-the manuscript uses the Δ ("del") symbol to indicate the transposon mutants, but this symbol is usually reserved for true genetic deletions, not transposon insertions. The strains should be denoted by the name of the allele, or another accepted indicator of Tn insertion. **We thank the reviewer for their comment and have made the corrections.**

-line 124: A polar effect typically involves reductions in a co-transcribed gene that is promoter-distal to the mutation. As diagrammed, however, the gene encoding TBDR2 is upstream of the gene encoding AbSLP, so polarity of AbSLP mutation on TBDR2 expression (as suggested in the text) appears to be an unlikely explanation. Polar effects of mutations on promoter-distal genes is a more likely scenario and could be ruled out as described above. **This has been addressed in lines 161-174.**

-Was the same exact WT data presented in each panel of Fig. 3d-i? If so this should be clearly indicated. **We have made the correction. Data in this figure is now in Supplementary Fig. 12, and it is noted that the same WT data is used in all the panels for ease of comparison.**

-In the in vitro growth experiments, the data presented with linear y-axis show defects in overall growth yield related to heme acquisition mutants, but defects in logarithmic growth rate/doubling time are difficult to ascertain. Were these observed? **Defects in logarithmic growth rate were not generally observed.**

Reviewer #3 (Comments for the Author)

The manuscript by Bateman et al. investigates the means by which *Acinetobacter baumannii*, an emerging opportunistic, and frequently multi-drug resistant pathogen, utilizes heme as an iron source. A previous bioinformatic analysis (Antunes LC, Imperi F, Towner KJ, Visca P. *Res Microbiol.* 2011 Apr;162(3)) revealed that some *A. baumannii* isolates possess two putative heme uptake clusters, heme clusters 1 and heme cluster 2 (hemO cluster), and Bateman and colleagues work towards elucidating the individual contributions of these two systems to heme-iron utilization. Through use of mutants disrupted for key components of each system, the authors find that only heme uptake cluster 2 contributes to the iron-dependent growth of *A. baumannii* on heme and hemoproteins. In beginning to characterize the machinery involved in heme uptake through cluster 2, the authors focus on two hypothetical proteins, a proposed surface-anchored lipoprotein (abSLP) and an associated surface protein assembly modulator (AbSlam). Both abSLP and AbSlam are required for growth on heme/hemoproteins as a sole iron source, and heterologous expression of these two proteins, along with TonB and a TonB-dependent transporter (TBDT2), is sufficient to facilitate heme utilization by *Escherichia coli*. In *E. coli*, abSLP-FLAG is found to be cell surface associated through labelling with an α -FLAG conjugated fluorophore and subsequent fluorescence detection. Following purification, the crystal structure of AbSLP was solved and essential residues in heme binding (H30 and H93) were identified. Lastly, the authors show that the TBDTs of both heme uptake systems are required for maximal survival and virulence of the bacteria in vivo, and that AbSlam, and to

a lesser extent, AbSLP also contribute to the pathogenesis of *A. baumannii*. Similar to iron acquisition systems in *Neisseria*, the authors propose a model where abSlam facilitates surface presentation of AbSLP, which in turn increases the efficiency of heme extraction and capture. Following capture, heme is then imported by TBDT2 and TonB for utilization by the bacteria.

Overall, the manuscript is well-written, flows logically, and is pleasant to read. Given the recent interest by both the WHO and CDC in better understanding the basic biology of *A. baumannii*, the paper is topical. However, concerns about novelty are raised by the fact that similar findings were recently published by another group (Giardina BJ, Shahzad S, Huang W, Wilks A. 2019. *Arch Biochem Biophys.* Sep 15;672:108066), identifying AbSLP as a hemophore that facilitates heme acquisition by the bacterium. At the very least, the authors need to reframe their findings in the context of the preexisting literature. Additional questions, concerns, and suggestions to help improve the paper are below.

Major criticisms/corrections:

1. As noted above, the role of heme uptake clusters 1 and 2 in heme acquisition by *Acinetobacter baumannii* was recently addressed in a manuscript by the Wilks Lab, where heme uptake cluster 2 (referred to as the hemO cluster) was found to be required for the heme-dependent growth of *A. baumannii* whilst heme cluster 1 was found to be dispensable for this function. Figure 1 of the current manuscript effectively recapitulates figures 1 and 2 of the aforementioned manuscript, albeit with a cleaner experimental design. Further, Giardina et al perform hemin-agarose pulldowns on the iron-restricted spent culture supernatants of *A. baumannii* strains AB0057 and LAC-4, and identify homologues of AbSLP (ABLAC_16810/AB57_0988). The authors name these proteins AbHph for their heme-binding properties and propose, given the AbSLP-homologues were identified in the extracellular fraction that they function not as surface-anchored lipoproteins but rather as secreted hemophores. The authors need to cite Giardina et al. discuss how their findings fit into the context of what is already known about *A. baumannii* heme acquisition, and address the discrepancy in proposed function. The fact that apo-abSLP in combination with hemoglobin can restore growth of Slam and SLP mutants indicates that the bacteria is capable of utilizing abSLP as a hemophore. Expression of tagged abSLP in *A. baumannii* Δ abSLP followed by fractionation and Western Blotting would help identify the localization of abSLP in its native bacteria, instead of in a heterologous expression system.

The *E. coli* platform is an artificial system to test whether AbHph (HphA) is a Slam dependent substrate. Although present on the cell surface, HphA has its endogenous signal peptide replaced with pelB signal sequence followed by the predicted HphA lipobox motif. This was done to ensure efficient Sec targeting in *E. coli*. We confirmed that HphA is secreted in the native organism, and this secretion is a Slam dependent process. We have re-written the article to address how our results fit in the context of the Giardina et al paper. However, our article is still novel because of the in-depth characterization of a novel Slam substrate.

2. Throughout the paper, common bacterial nomenclature is not adhered to in the naming of AbSLP, AbSLM, TBDR1, and TBDR2. Please rename so that the genes/proteins fit the convention of three lower case letters indicating the pathway/process and an uppercase

letter specifying the gene/gene product. Given the AbSLP has already been named AbHph by the aforementioned group, it might be appropriate to incorporate part of this nomenclature in the new name. **The re-submitted manuscript contains new names.**

3. An author on the current manuscript has previously published on the potential role of the hemO cluster on *A. baumannii* virulence, including bioinformatic analyses on the conservation of the hemO cluster in clinical isolates (Ou et al. (2015) *Sci Rep* 5(8643); de Léséleuc (2014) *Int J Med Microbiol* 304(3-4)). It is unclear why these papers were not cited as part of the background literature/rationale for the study (the former appears to be cited only as a strain reference, and the latter does not appear to be addressed at all). **We thank the reviewer for their comment, and have included these references as background.**

4. The authors make mention of the domain architectures of the SLP and Slam proteins; it would be helpful to have an actual schematic detailing this information in the supplement. **We have added this information in Supplementary Figure 3.**

5. It is evident that AbSLP binds heme, and the authors suggest that the protein is also capable of binding both holo and apo-hemoglobin and that the same residues (H30 and H93) participate in these interactions. Heme-binding blocks the interaction with hemoglobin, which suggests heme and hemoglobin are bound at the same location. How do the authors propose that apo-hemoglobin is bound, if heme is not participating in the coordination? **We have updated residue numbering such that H30 and H93 are now H43 and H106 respectively. H30 is located on a flexible loop in the crystal structure, and this loop may wedge itself into the heme binding pocket. We propose that heme spontaneously released from Hb is first bound by H106 (fast step). This causes the loop containing H43 to be released from Hb and enclose onto the heme (slow step). This biphasic mode of heme binding is plausible and support comes from studies on the HasA hemophore, which also binds to heme using a flexible loop containing a His residue that moves 20-30Å to clamp down on heme. This model is mentioned in lines 218-230 of the discussion.**

6. In preparing the inoculum for the mouse infections, it appears that the bacteria were iron-starved first. Firstly, this should be noted in the main text, as it is not a common procedure in preparing bacteria for murine models of bacterial infection. Secondly, was this a necessary measure to obtain the phenotypes observed? It is possible that the mutants are already severely attenuated in vitro before being introduced to the mouse? Do the strains grow comparably under these conditions?

We starved the cells of iron to deplete their iron stores and force them to acquire iron from the environment. These strains do not appear to be attenuated prior to inoculating mice. This data was added in Supplementary Figure 13.

Minor criticisms/corrections:

Title: Capitalization is irregular (e.g. Heme is capitalized by acquisition isn't). **We have made the correction.**

Title: Italicize *Acinetobacter*. **We have made the correction.**

Page 2, line 3: Here *Acinetobacter baumannii* is referred to as a nosocomial pathogen, but in the first line it is mentioned that it causes both hospital and community acquired infections. Perhaps for consistency "opportunistic pathogen" would be more appropriate in this

context. **We thank the reviewer for their comment and have made the correction.**

Page 2, line 12: Dissemination into the vasculature is not assessed directly. The lack of Δ TBDR2 bacteria in the blood could be due to survival and/or dissemination defects. **We thank the reviewer and have made the clarification.**

Page 2, line 18: Delete “disease in”. **We have made the change.**

Page 2, line 20: Define the acronym “UTIs”. **We have made the change.**

Page 3, line 28: This sentence is a bit confusing. Perhaps rephrase to something suggesting that *A. baumannii*’s demand for iron could potentially be exploited as a weakness. **We thank the reviewer and have made the clarification.**

Page 3, line 43: What kind of closer inspection? Presumably these proteins bear homology to neisserial SLP and Slam proteins and were identified through bioinformatics in the authors’ earlier Nature Microbiology paper? **We thank the reviewer and have elaborated on this section to make it clearer (lines 54-64).**

Page 3, line 44: How are they misannotated? What were they annotated as prior? **We thank the reviewer and have elaborated on this section to make it clearer (lines 54-64).**

Page 5, line 72: Which TonB? Most *A. baumannii* strains express several. **We have included in brackets the name proper name of the TonB, which is ABUW_2982.**

Page 5, line 83: Delete “reconstitution assay.” **We have made the change.**

Page 6, Line 92: Switch α to alpha, or beta to β for consistency. **We have made the change.**

Page 6, line 105: Add word “chromatography” and either capitalize all of the words or none of the words in Size Exclusion Chromatography with Multi-Angle Light Scattering. **We have made the change.**

Page 8, line 137: Change “Figure 4e-h” to “Figure 4e-g,” as there is no panel “h”. **We have corrected this.**

Page 9, line 180: The authors discuss anchoring and subsequent cleavage of HmuY from the cell surface, where it then functions as a hemophore. As stated above, it is possible that the same is true of AbSLP. **There is no evidence to suggest that HphA is lipidated like HmuY. Although we do not know for sure the full HphA translocation pathway, the signal peptide is predicted to be processed by Signal Peptidase I, which belongs to the soluble secretion pathway as opposed to Signal Peptidase II, which processes immature lipoproteins.**

Page 10, first paragraph: Given that heme cluster 1 lacks a heme oxygenase, unless one is encoded elsewhere in the genome, it is possible this cluster does not facilitate iron acquisition from heme (this is true of *A. baumannii* ATCC 17978). **This is addressed in lines 363-371.**

Page 10, final paragraph: The authors suggest that the Slam containing heme acquisition operon (heme cluster 2, hemO cluster) makes *A. baumannii* more virulent in a mouse model and highlight the possibility that this cluster might be conserved in hospital isolates. Giardina et al found that ~60% of clinical isolates possess the hemO cluster. **We have included this in the discussion.**

Page 13, line 268 and page 14, line 277: Change “kDA” to “kDa” **We have made the change.**

Figure 1 caption: Add “(HSA)” after “human serum albumin”. **We have made the correction.**

Figure 1 caption: Add a space between “600” and “nm.” **We have made the correction.**

Figure 2 caption: Define “PE.” **We have added the full name.**

Figure 3 caption: Change “The plotted curves represents” to “represent” **We have made the correction.**

Supplementary Figure 4: Shouldn’t the MW of the holo protein be greater than the apo

protein (see inset table)? It is possible that some aggregation or “stickiness” is occurring with the apo protein, which could explain the larger MW for the apo vs holo protein.

Supplementary Figure 4: Change kDA to kDa in y-axis label. We have made the correction.

REVIEWERS' COMMENTS

Reviewer #1 (Remarks to the Author):

[No comments for authors]

Reviewer #2 (Remarks to the Author):

The resubmitted manuscript provides new data that largely addresses my concerns. The new complementation experiments (Fig. S11) show that all of the Tn mutants analyzed (hphA, hphR, hsmA) show some degree of non-complementation/polarity effects. These findings are adequately described, but they introduce a major caveat in the animal experiments that should be addressed with more attention in the text. That is, when describing results with the mutants in the animal model, it should be emphasized that each mutant has the observed polar/extended effects on other elements of the system, and the results should be interpreted in that context.

Overall, the resubmission is much improved. The added discussion text is excellent, thorough, and makes it easy to place the work in context with other research on heme/iron acquisition systems.

I had the following minor comments:

-line 91-92 in the statement, "enables *E. coli* to use hemoglobin as an iron source without conferring a general growth advantage" should be rephrased to make it clear that "general growth advantage" refers to growth in rich bacteriological medium as opposed to the Hb minimal medium.

-line 593, line 871: In "Pull Down with AbHph-GST Coated Resin", and "SDS-PAGE gel of Hpx and purified HphA-GST (~50 kDa) is shown to the right", should state explicitly how proteins were detected after SDS-PAGE.

-Fig. S10A change AbHph to HphA

-line 166: "hampers growth in the presence of purified holo HphA". The error bars are large with these data points -- to justify this statement, statistics should be performed (such as the analysis done in 12 d-f); if not significant, the statement should be toned down/reworded.

-lines 171-175: as the data are presented in this text, please refer to the specific figure panels showing the data.

-line 417: should read "Cre recombinase mediated excision of the antibiotic marker," or similar, rather than "the transposon"

-line 483: anti-GroEL western blotting is described, but this analysis is not shown in the corresponding figure; would be a useful loading control to show.

-I would also consider showing elements of Figs. S10-12 as part of the main figures.

Reviewer #3 (Remarks to the Author):

In the resubmission of "Slam Dependent Hemophore Aids in Heme Acquisition For The Bacterial Pathogen, *Acinetobacter baumannii*," Bateman *et al*. provide a comprehensive evaluation of the heme acquisition pathways of this emerging pathogen. HphA, a recently discovered hemophore (Giardina BJ, Shahzad S, Huang W, Wilks A. 2019. Arch Biochem Biophys. Sep 15;672:108066) is shown to be a novel substrate of the Slam translocation machinery of *A. baumannii* (HsmA). The *in vitro* contribution of heme cluster 1 (*hemRT*) and heme cluster 2 (*hphA*, *hphR*, and *hsmA*) to heme-iron acquisition are evaluated and heme cluster 2 but not heme cluster 1 is found to be required to

support growth on heme/hemoglobin under iron-restriction. HphA is shown to be a *bona fide* heme-binding protein, that interacts with hemoglobin and passively acquires iron from heme. Structural elucidation of HphA reveals homology to neisserial surface lipoproteins and to a hemophore of *Haemophilus haemolyticus*. The contribution of both heme cluster 1 and heme cluster 2 are assessed for their contribution to *A. baumannii* survival and pathogenesis in vivo, where although a *hemRT* mutant is somewhat attenuated in a murine model of pneumonia, an *hphR* mutant (disrupted for the expression of a TonB dependent receptor protein) and to a lesser extent an *hsmA* mutant are strongly attenuated in this model. Both *hphA* and *hphR* are required for maximal virulence in a murine model of sepsis. The authors have sufficiently addressed the concerns I raised in the first submission. Although I have made some minor editorial suggestions below, I have no major comments or concerns with the manuscript. I believe that this manuscript will help provide understanding into the basic biology of *A. baumannii* and the mechanisms utilized to survive within the host which is sorely lacking in the current literature.

General comments:

- 1) Some of the previous nomenclature for the gene names is still present in the figures.
- 2) The flow of the discussion is a bit rough in some places with newly added sections, but I will leave it to the discretion of the authors to polish it further.

Minor corrections/suggestions:

Page 6, lines 102-103. Is supplementary figure 3 the correct figure for this statement? It shows the domain architecture of HsmA and HphA, not an *E. coli* assay.

Page 6, line 109. It would make more sense for the figures to be discussed in order.

Page 7, line 129. The crystallization data is shown in panels c and d of Supplementary Figure 6, not a and b, which are UV/vis spectra.

Page 10, line 188 and 192. The figure shows burden data in panels b, c, and d, and clinical scoring in e, f, and g. This is reversed in the text and figure legend (page 40 lines 877-886).

Page 14, line 286. Does *Haemophilus haemolyticus* have a Slam protein homologue?

Page 14, line 291-292. This sentence could use some clarification, since one would not anticipate large amounts of heme during colonization of the respiratory tract.

Page 15, line 316. It has been shown in several studies that *A. baumannii* ATCC 17978 can cause bacteraemia in murine models of infection (including papers cited on page 16 of this manuscript), so this is perhaps not the best way to delineate LAC-4 from 17978.

Page 15, line 320. The authors state that that "both LAC-4 and 17978 are similar in their ability to produce iron chelating siderophores, and inability to use transferrin and lactoferrin." This might be a typo, but *A. baumannii* can use both transferrin and lactoferrin as iron sources.

Page 16, line 327. Change "clusters" to "cluster".

Page 30-37. Genera and species of bacteria are not italicized in the reference list.

Page 39, line 847. Add a space between "on" and "HsmA"

Figure 1a. If these are to represent genetic loci, the gene names should be italicized and the first letters not capitalized.

Figure 3b. Change labeling on y-axis to HphA instead of AbHphA.

Supplementary figure 6. I may have missed it, but I don't believe the UV/vis spectra were discussed in the text, what are we supposed to be seeing here? The spectra are very small and hard to see.

Supplementary figure 10a. Graph is labeled Holo AbHph instead of Holo HphA.

Supplementary figure 11a-c. The growth curves are quite tiny and hard to read the labeling.

Supplementary strains table. A description for *A. baumannii* strains LAC4 and AB5075 are missing. Where/when were these isolated? What is their drug resistance profile? Any other notable attributes?

Supplementary references. *Acinetobacter baumannii* is not italicized in the references.

Responses to Reviewer Comments:

We thank the reviewers for their helpful comments and feedback. Reviewer comments are listed below with their respective responses and we have included line numbers or figures for relevant sections of the manuscript (the line numbers are in reference to the version of the word doc with track changes showing) .

Reviewer #1 (Remarks to the Author)

[No comments for authors]

Reviewer #2 (Remarks to the Author)

The resubmitted manuscript provides new data that largely addresses my concerns. The new complementation experiments (Fig. S11) show that all of the Tn mutants analyzed (hphA, hphR, hsmA) show some degree of non-complementation/polarity effects. These findings are adequately described, but they introduce a major caveat in the animal experiments that should be addressed with more attention in the text. That is, when describing results with the mutants in the animal model, it should be emphasized that each mutant has the observed polar/extended effects on other elements of the system, and the results should be interpreted in that context. **We have made this distinction on page 9 with reference to supplementary figure 10a-c.**

Overall, the resubmission is much improved. The added discussion text is excellent, thorough, and makes it easy to place the work in context with other research on heme/iron acquisition systems.

I had the following minor comments:

-line 91-92 in the statement, "enables E. coli to use hemoglobin as an iron source without conferring a general growth advantage" should be rephrased to make it clear that "general growth advantage" refers to growth in rich bacteriological medium as opposed to the Hb minimal medium. **We have clarified this.**

-line 593, line 871: In "Pull Down with AbHph-GST Coated Resin", and "SDS-PAGE gel of Hpx and purified HphA-GST (~50 kDa) is shown to the right", should state explicitly how proteins were detected after SDS-PAGE. **We have added this information.**

-Fig. S10A change AbHph to HphA. **We have made the change.**

-line 166: "hampers growth in the presence of purified holo HphA". The error bars are large

with these data points -- to justify this statement, statistics should be performed (such as the analysis done in 12 d-f); if not significant, the statement should be toned down/reworded.

The difference between empty vector and plasmid complemented HphA mutant grown in the presence of holo HphA is not statistically significant so we toned down the statement.

-lines 171-175: as the data are presented in this text, please refer to the specific figure panels showing the data. We have made the change.

-line 417: should read "Cre recombinase mediated excision of the antibiotic marker," or similar, rather than "the transposon" We have made the change.

-line 483: anti-GroEL western blotting is described, but this analysis is not shown in the corresponding figure; would be a useful loading control to show. The anti-GroEL control for cell lysis is shown in Figure 2a (bottom blot).

-I would also consider showing elements of Figs. S10-12 as part of the main figures. We have incorporated Figs. S10 and 12 as main figures.

Reviewer #3 (Remarks to the Author):

In the resubmission of "Slam Dependent Hemophore Aids in Heme Acquisition For The Bacterial Pathogen, *Acinetobacter baumannii*," Bateman *et al.* provide a comprehensive evaluation of the heme acquisition pathways of this emerging pathogen. HphA, a recently discovered hemophore (Giardina BJ, Shahzad S, Huang W, Wilks A. 2019. Arch Biochem Biophys. Sep 15;672:108066) is shown to be a novel substrate of the Slam translocation machinery of *A. baumannii* (HsmA). The *in vitro* contribution of heme cluster 1 (*hemRT*) and heme cluster 2 (*hphA*, *hphR*, and *hsmA*) to heme-iron acquisition are evaluated and heme cluster 2 but not heme cluster 1 is found to be required to support growth on heme/hemoglobin under iron-restriction. HphA is shown to be a *bona fide* heme-binding protein, that interacts with hemoglobin and passively acquires iron from heme. Structural elucidation of HphA reveals homology to neisserial surface lipoproteins and to a hemophore of *Haemophilus haemolyticus*. The contribution of both heme cluster 1 and heme cluster 2 are assessed for their contribution to *A. baumannii* survival and pathogenesis *in vivo*, where although a *hemRT* mutant is somewhat attenuated in a murine model of pneumonia, an *hphR* mutant (disrupted for the expression of a TonB dependent receptor protein) and to a lesser extent an *hsmA* mutant are strongly attenuated in this model. Both *hphA* and *hphR* are required for maximal virulence in a murine model of sepsis. The authors have sufficiently addressed the concerns I raised in the first submission. Although I have made some minor editorial suggestions below, I have no major comments or concerns with the manuscript. I believe that this manuscript will help provide understanding into the basic biology of *A. baumannii* and the mechanisms utilized to survive within the host which is sorely lacking in the current literature.

General comments:

1) Some of the previous nomenclature for the gene names is still present in the figures.

Previous names have been removed or renamed.

2) The flow of the discussion is a bit rough in some places with newly added sections, but I will leave it to the discretion of the authors to polish it further. We have added transition sentences to make the discussion flow better.

Minor corrections/suggestions:

Page 6, lines 102-103. Is supplementary figure 3 the correct figure for this statement? It shows the domain architecture of HsmA and HphA, not an *E. coli* assay. We have made the change.

Page 6, line 109. It would make more sense for the figures to be discussed in order. We have made this discussion clearer.

Page 7, line 129. The crystallization data is shown in panels c and d of Supplementary Figure 6, not a and b, which are UV/vis spectra. We have made the change. We refer to both the UV-VIS data and structure as support for the absence of heme in apo HphA (lines 166-168)

Page 10, line 188 and 192. The figure shows burden data in panels b, c, and d, and clinical scoring in e, f, and g. This is reversed in the text and figure legend (page 40 lines 877-886). We have made the change.

Page 14, line 286. Does *Haemophilus haemolyticus* have a Slam protein homologue? Yes, and this information is now included (line 385).

Page 14, line 291-292. This sentence could use some clarification, since one would not anticipate large amounts of heme during colonization of the respiratory tract. We have clarified this idea (lines 388-392).

Page 15, line 316. It has been shown in several studies that *A. baumannii* ATCC 17978 can cause bacteraemia in murine models of infection (including papers cited on page 16 of this manuscript), so this is perhaps not the best way to delineate LAC-4 from 17978. ATCC 17978 is considered a non-bacteremic strain in immunocompetent mice according to de Léséleuc *et al.* (2014) *Int. J. Med. Microbiol.* **304**, 360-369, which we have now highlighted in lines 423-424. This particular study compared ATCC 17978 to the strain, LAC-4, which can invade the bloodstream after colonizing the nasopharynx. Other *A. baumannii* bacteremic infection models are the result of injecting bacteria directly into the circulation of mice as opposed to first developing pneumonia and then progressing to bacteremia.

Page 15, line 320. The authors state that that “both LAC-4 and 17978 are similar in their ability to produce iron chelating siderophores, and inability to use transferrin and lactoferrin.” This might be a typo, but *A. baumannii* can use both transferrin and lactoferrin as iron sources. We have made the change.

Page 16, line 327. Change “clusters” to “cluster”. We have made the change.

Page 30-37. Genera and species of bacteria are not italicized in the reference list. We have made the change.

Page 39, line 847. Add a space between “on” and “HsmA” We have made the change.

Figure 1a. If these are to represent genetic loci, the gene names should be italicized and the first letters not capitalized. We have now indicated that these are protein annotations and not genes.

Figure 3b. Change labeling on y-axis to HphA instead of AbHphA. **We have made the change.**

Supplementary figure 6. I may have missed it, but I don't believe the UV/vis spectra were discussed in the text, what are we supposed to be seeing here? The spectra are very small and hard to see. **The UV-Vis spectra are now clearly referenced and provide further confirmation that apo HphA is not bound to heme (lines 166-168). Spectra are now enlarged.**

Supplementary figure 10a. Graph is labeled Holo AbHph instead of Holo HphA. **We have made the change.**

Supplementary figure 11a-c. The growth curves are quite tiny and hard to read the labeling. **Graphs are now enlarged.**

Supplementary strains table. A description for *A. baumannii* strains LAC4 and AB5075 are missing. Where/when were these isolated? What is their drug resistance profile? Any other notable attributes? **A thorough description of the strains are now included.**

Supplementary references. *Acinetobacter baumannii* is not italicized in the references. **We have made the change.**